# FAST NONLINEAR VECTOR QUANTILE REGRESSION

**Aviv A. Rosenberg**[1,3,†], **Sanketh Vedula**[1,3,†], **Yaniv Romano**[1,2], and **Alex M. Bronstein**[1,3]

[1]Department of Computer Science, Technion
[2]Department of Electrical and Computer Engineering, Technion
[3]Sibylla, UK
{avivr,sanketh,yromano,bron}@cs.technion.ac.il
[†]Equal Contribution

## ABSTRACT

Quantile regression (QR) is a powerful tool for estimating one or more conditional quantiles of a target variable Y given explanatory features **X**. A limitation of QR is that it is only defined for scalar target variables, due to the formulation of its objective function, and since the notion of quantiles has no standard definition for multivariate distributions. Recently, vector quantile regression (VQR) was proposed as an extension of QR for vector-valued target variables, thanks to a meaningful generalization of the notion of quantiles to multivariate distributions via optimal transport. Despite its elegance, VQR is arguably not applicable in practice due to several limitations: (i) it assumes a linear model for the quantiles of the target **Y** given the features **X**; (ii) its exact formulation is intractable even for modestly-sized problems in terms of target dimensions, number of regressed quantile levels, or number of features, and its relaxed dual formulation may violate the monotonicity of the estimated quantiles; (iii) no fast or scalable solvers for VQR currently exist. In this work we fully address these limitations, namely: (i) We extend VQR to the non-linear case, showing substantial improvement over linear VQR; (ii) We propose vector monotone rearrangement, a method which ensures the quantile functions estimated by VQR are monotone functions; (iii) We provide fast, GPU-accelerated solvers for linear and nonlinear VQR which maintain a fixed memory footprint, and demonstrate that they scale to millions of samples and thousands of quantile levels; (iv) We release an optimized python package of our solvers as to widespread the use of VQR in real-world applications.

## 1 INTRODUCTION

Quantile regression (QR) (Koenker & Bassett, 1978) is a well-known method which estimates a *conditional quantile* of a target variable Y, given covariates **X**. A major limitation of QR is that it deals with a scalar-valued target variable, while many important applications require estimation of vector-valued responses. A trivial approach is to estimate conditional quantiles separately for each component of the vector-valued target. However this assumes statistical independence between targets, a very strong assumption rarely held in practice. Extending QR to high dimensional responses is not straightforward because (i) the notion of quantiles is not trivial to define for high dimensional variables, and in fact multiple definitions of multivariate quantiles exist (Carlier et al., 2016); (ii) quantile regression is performed by minimizing the *pinball loss*, which is not defined for high dimensional responses.

Seminal works of Carlier et al. (2016) and Chernozhukov et al. (2017) introduced a notion of quantiles for vector-valued random variables, termed *vector quantiles*. Key to their approach is extending the notions of monotonicity and strong representation of scalar quantile functions to high dimensions, i.e.

$$\text{\textit{Co-monotonicity}}: \left(Q_{\mathbf{Y}}(\boldsymbol{u}) - Q_{\mathbf{Y}}(\boldsymbol{u}')\right)^{\top}\left(\boldsymbol{u} - \boldsymbol{u}'\right) \geq 0, \ \forall \, \boldsymbol{u}, \boldsymbol{u}' \in [0,1]^d \tag{1}$$

$$\text{\textit{Strong representation}}: \mathbf{Y} = Q_{\mathbf{Y}}(\mathbf{U}), \ \mathbf{U} \sim \mathbb{U}[0,1]^d \tag{2}$$

where **Y** is a $d$-dimensional variable, and $Q_{\mathbf{Y}} : [0,1]^d \mapsto \mathbb{R}^d$ is its *vector quantile function* (VQF).

Moreover, Carlier et al. (2016) extended QR to vector-valued targets, which leads to *vector quantile regression* (VQR). VQR estimates the *conditional vector quantile function* (CVQF) $Q_{\mathbf{Y}|\mathbf{X}}$ from

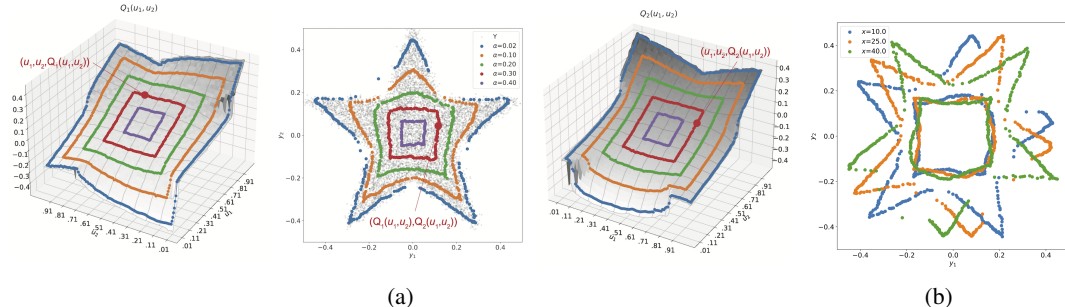

(a)                                                                    (b)

Figure 1: **(a) Visualization of the vector quantile function (VQF) and its $\alpha$-contours, a high-dimensional generalization of $\alpha$-confidence intervals.** Data was drawn from a 2d star-shaped distribution. Vector quantiles (colored dots) are overlaid on the data (middle). Different colors correspond to $\alpha$-contours, each containing $100 \cdot (1 - 2\alpha)^2$ percent of the data. The VQF $Q_{\mathbf{Y}}(\boldsymbol{u}) = [Q_1(\boldsymbol{u}), Q_2(\boldsymbol{u})]^\top$ is co-monotonic with $\boldsymbol{u} = (u_1, u_2)$; $Q_1, Q_2$ are depicted as surfaces (left, right) with the corresponding vector quantiles overlaid. On $Q_1$, increasing $u_1$ for a fixed $u_2$ produces a monotonically increasing curve, and vice versa for $Q_2$. **(b) Visualization of conditional vector quantile functions (CVQFs) via $\alpha$-contours.** Data was drawn from a joint distribution of $(\mathbf{X}, \mathbf{Y})$ where $\mathbf{Y}|\mathbf{X} = \boldsymbol{x}$ has a star-shaped distribution rotated by $\boldsymbol{x}$ degrees. The true CVQF $Q_{\mathbf{Y}|\mathbf{X}}$ changes non-linearly with the covariates $\mathbf{X}$, while $\mathbb{E}[\mathbf{Y}|\mathbf{X}]$ remains the same. This demonstrates the challenge of estimating CVQFs from samples of the joint distribution. Appendix C provides further intuitions regarding VQFs and CVQFs, and details how the $\alpha$-contours are constructed from them.

samples drawn from $P_{(\mathbf{X}, \mathbf{Y})}$, where $\mathbf{Y}$ is a $d$-dimensional target variable and $\mathbf{X}$ are $k$-dimensional covariates. They show that a function $Q_{\mathbf{Y}|\mathbf{X}}$ which obeys co-monotonicity (1) and strong representation (2) exists and is unique, as a consequence of Brenier's polar factorization theorem (Brenier, 1991). Figure 1 provides a visualization of these notions for a two-dimensional target variable.

Assuming a linear specification $Q_{\mathbf{Y}|\mathbf{X}}(\boldsymbol{u}; \boldsymbol{x}) = \boldsymbol{B}(\boldsymbol{u})^\top \boldsymbol{x} + \boldsymbol{a}(\boldsymbol{u})$, VQR can be formulated as an optimal transport problem between the measures of $\mathbf{Y}|\mathbf{X}$ and $\mathbf{U}$, with the additional mean-independence constraint $\mathbb{E}[\mathbf{U}|\mathbf{X}] = \mathbb{E}[\mathbf{X}]$. The primal formulation of this problem is a large-scale linear program and is thus intractable for modestly-sized problems. A relaxed dual formulation which is amenable to gradient-based solvers exists but leads to co-monotonicity violations.

The *first goal* of our work is to address the following limitations of Carlier et al. (2016; 2020): (i) the linear specification assumption on the CVQF, and (ii) the violation of co-monotonicity when solving the inexact formulation of the VQR problem. The *second goal* of this work is to make VQR an accessible tool for off-the-shelf usage on large-scale high-dimensional datasets. Currently there are no available software packages to estimate VQFs and CVQFs that can scale beyond toy problems. We aim to provide accurate, fast and distribution-free estimation of these fundamental statistical quantities. This is relevant for innumerable applications requiring statistical inference, such as distribution-free uncertainty estimation for vector-valued variables (Feldman et al., 2021), hypothesis testing with a vector-valued test statistic (Shi et al., 2022), causal inference with multiple interventions (Williams & Crespi, 2020), outlier detection (Zhao et al., 2019) and others. Below we list our contributions.

**Scalable VQR.** We introduce a highly-scalable solver for VQR in Section 3. Our approach, inspired by Genevay et al. (2016) and Carlier et al. (2020), relies on solving a new relaxed dual formulation of the VQR problem. We propose custom stochastic-gradient-based solvers which maintain a constant memory footprint regardless of problem size. We demonstrate that our approach scales to millions of samples and thousands of quantile levels and allows for GPU-acceleration.

**Nonlinear VQR.** To address the limitation of linear specification, in Section 4 we propose *nonlinear vector quantile regression* (NL-VQR). The key idea is fit a nonlinear embedding function of the input features jointly with the regression coefficients. This is made possible by leveraging the relaxed dual formulation and solver introduced in Section 3. We demonstrate, through synthetic and real-data experiments, that nonlinear VQR can model complex conditional quantile functions substantially better than linear VQR and separable QR approaches.

**Vector monotone rearrangement (VMR).** In Section 5 we propose VMR, which resolves the co-monotonicity violations in estimated CVQFs. We solve an optimal transport problem to *rearrange*

the vector quantiles such that they satisfy co-monotonicity. We show that VMR strictly improves the estimation quality of CVQFs while ensuring zero co-monotonicity violations.

**Open-source software package.** We release a feature-rich, well-tested python package vqr[1], implementing estimation of vector quantiles, vector ranks, vector quantile contours, linear and nonlinear VQR, and VMR. To the best of our knowledge, this would be the first publicly available tool for estimating conditional vector quantile functions at scale.

## 2 BACKGROUND

Below we introduce quantile regression, its optimal-transport-based formulation and the extension to vector-valued targets. We aim to provide a brief and self-contained introduction to the key building blocks of VQR. For simplicity, we present the discretized versions of the problems.

**Notation.** Throughout, Y, $\mathbf{X}$ denote random variables and vectors, respectively; deterministic scalars, vectors and matrices are denoted as $y$, $\boldsymbol{x}$, and $\boldsymbol{X}$. $P_{(\mathbf{X},\mathrm{Y})}$ denotes the joint distribution of the $\mathbf{X}$ and Y. $\mathbf{1}_N$ denotes an $N$-dimensional vector of ones, $\odot$ denotes the elementwise product, and $\mathbb{I}\{\cdot\}$ is an indicator. We denote by $N$ the number of samples, $d$ the dimension of the target variable, $k$ the dimension of the covariates, and $T$ the number of vector quantile levels per target dimension. $Q_{\mathbf{Y}|\mathbf{X}}(\boldsymbol{u};\boldsymbol{x})$ is the CVQF of the variable $\mathbf{Y}|\mathbf{X}$ evaluated at the vector quantile level $\boldsymbol{u}$ for $\mathbf{X}=\boldsymbol{x}$.

**Quantile Regression.** The goal of QR is to estimate the quantile of a variable Y$|\mathbf{X}$. Assuming a linear model of the latter, and given $u \in (0,1)$, QR amounts to solving $(\boldsymbol{b}_u, a_u) = \arg\min_{\boldsymbol{b},a} \mathbb{E}_{(\mathbf{X},\mathrm{Y})}\left[\rho_u(\mathrm{Y} - \boldsymbol{b}^\top \mathbf{X} - a)\right]$, where $\boldsymbol{b}_u{}^\top \boldsymbol{x} + a_u$ is the $u$-th quantile of Y$|\mathbf{X}=\boldsymbol{x}$, and $\rho_u(z)$, known as the *pinball loss*, is given by $\rho_u(z) = \max\{0, z\} + (u-1)z$. Solving this problem produces an estimate of $Q_{\mathrm{Y}|\mathbf{X}}$ for a single quantile level $u$. In order to estimate the full *conditional quantile function* (CQF) $Q_{\mathrm{Y}|\mathbf{X}}(u)$, the problem must be solved at all levels of $u$ with additional monotonicity constraints, since the quantile function is non-decreasing in $u$. The CQF discretized at $T$ quantile levels can be estimated from $N$ samples $\{\boldsymbol{x}_i, y_i\}_{i=1}^N \sim P_{(\mathbf{X},\mathrm{Y})}$ by solving

$$\min_{\boldsymbol{B},\boldsymbol{a}} \ \sum_u \sum_{i=1}^N \rho_u(y_i - \boldsymbol{b}_u{}^\top \boldsymbol{x}_i - a_u) \tag{3}$$
$$\text{s.t. } \forall i, \ u' \geq u \implies \boldsymbol{b}_{u'}{}^\top \boldsymbol{x}_i + a_{u'} \geq \boldsymbol{b}_u{}^\top \boldsymbol{x}_i + a_u,$$

where $\boldsymbol{B}$ and $\boldsymbol{a}$ aggregate all the $\boldsymbol{b}_u$ and $a_u$, respectively. We refer to eq. (3) as *simultaneous linear quantile regression* (SLQR). This problem is undefined for a vector-valued Y, due to the inherently 1d formulation of the monotonicity constraints and the pinball loss $\rho_u(z)$.

**Optimal Transport Formulation.** Carlier et al. (2016) showed that SLQR (3) can be equivalently written as an *optimal transport* (OT) problem between the target variable and the quantile levels, with an additional constraint of mean independence. Given $N$ data samples arranged as $\boldsymbol{y} \in \mathbb{R}^N$, $\boldsymbol{X} \in \mathbb{R}^{N \times k}$, and $T$ quantile levels denoted by $\boldsymbol{u} = \left[\frac{1}{T}, \frac{2}{T}, ..., 1\right]^\top$ we can write,

$$\max_{\boldsymbol{\Pi} \geq 0} \ \boldsymbol{u}^\top \boldsymbol{\Pi} \boldsymbol{y}$$
$$\text{s.t. } \boldsymbol{\Pi}^\top \mathbf{1}_T = \boldsymbol{\nu}$$
$$\boldsymbol{\Pi} \mathbf{1}_N = \boldsymbol{\mu} \qquad [\boldsymbol{\varphi}] \tag{4}$$
$$\boldsymbol{\Pi} \boldsymbol{X} = \bar{\boldsymbol{X}} \qquad [\boldsymbol{\beta}]$$

where $\boldsymbol{\Pi}$ is the transport plan between quantile levels $\boldsymbol{u}$ and samples $(\boldsymbol{x}, \boldsymbol{y})$, with marginal constraints $\boldsymbol{\nu} = \frac{1}{N}\mathbf{1}_N$, $\boldsymbol{\mu} = \frac{1}{T}\mathbf{1}_T$ and mean-independence constraint $\bar{\boldsymbol{X}} = \frac{1}{T}\mathbf{1}_T \frac{1}{N}\mathbf{1}_N{}^\top \boldsymbol{X}$. The dual variables are $\boldsymbol{\varphi} = \boldsymbol{D}^{-\top}\boldsymbol{a}$ and $\boldsymbol{\beta} = \boldsymbol{D}^{-\top}\boldsymbol{B}$, where $\boldsymbol{D}^\top$ is a first-order finite differences matrix, and $\boldsymbol{a} \in \mathbb{R}^T$, $\boldsymbol{B} \in \mathbb{R}^{T \times k}$ contain the regression coefficients for all quantile levels. Refer to appendices A.1 and A.2 for a full derivation of the connection between SLQR (3) and OT (4).

**Vector Quantile Regression.** Although the OT formulation for SLQR (4) is specified between 1d measures, this formulation is immediately extensible to higher dimensions. Given vector-valued targets $\boldsymbol{y}_i \in \mathbb{R}^d$ arranged in $\boldsymbol{Y} \in \mathbb{R}^{N \times d}$, their vector quantiles are also in $\mathbb{R}^d$. The vector quantile

---

[1]can be installed with `pip install vqr`; source available at `https://github.com/vistalab-technion/vqr`

levels are sampled on a uniform grid on $[0,1]^d$ with $T$ evenly spaced points in each dimension, resulting in $T^d$ $d$-dimensional vector quantile levels, arranged as $\boldsymbol{U} \in \mathbb{R}^{T^d \times d}$. The OT objective can be written as $\sum_{i=1}^{T} \sum_{j=1}^{N} \Pi_{i,j} u_i y_j = \boldsymbol{\Pi} \odot \boldsymbol{S}$, where $\boldsymbol{S} \in \mathbb{R}^{T \times N}$, and thus can be naturally extended to $d > 1$ by defining the pairwise inner-product matrix $\boldsymbol{S} = \boldsymbol{U}\boldsymbol{Y}^\top \in \mathbb{R}^{T^d \times N}$. The result is a $d$-dimensional discrete linear estimate of the CVQF, $\widehat{Q}_{\mathbf{Y}|\mathbf{X}}(\boldsymbol{u}; \boldsymbol{x})$ which is co-monotonic (1) with $\boldsymbol{u}$ for each $\boldsymbol{x}$. Appendix A.6 details how the estimated CVQF is obtained from the dual variables in the high-dimensional case.

## 3 SCALABLE VQR

To motivate our proposed scalable approach to VQR, we first present the exact formulations of VQR and discuss their limitations. The OT-based primal and dual problems are presented below.

$$
\begin{aligned}
\max_{\boldsymbol{\Pi} \geq 0} \quad & \sum_{i=1}^{T^d} \sum_{j=1}^{N} \boldsymbol{u}_i^\top \boldsymbol{y}_j \Pi_{i,j} \\
\text{s.t.} \quad & \boldsymbol{\Pi}^\top \mathbf{1}_{T^d} = \boldsymbol{\nu} \quad [\boldsymbol{\psi}] \\
& \boldsymbol{\Pi}\mathbf{1}_N = \boldsymbol{\mu} \quad [\boldsymbol{\varphi}] \\
& \boldsymbol{\Pi}\boldsymbol{X} = \bar{\boldsymbol{X}} \quad [\boldsymbol{\beta}]
\end{aligned} \tag{5}
$$

$$
\begin{aligned}
\min_{\boldsymbol{\psi},\boldsymbol{\varphi},\boldsymbol{\beta}} \quad & \boldsymbol{\psi}^\top \boldsymbol{\nu} + \boldsymbol{\varphi}^\top \boldsymbol{\mu} + \text{tr}\left(\boldsymbol{\beta}^\top \bar{\boldsymbol{X}}\right) \\
\text{s.t.} \quad & \forall i, j: \\
& \varphi_i + \boldsymbol{\beta}_i^\top \boldsymbol{x}_j + \psi_j \geq \boldsymbol{u}_i^\top \boldsymbol{y}_j \quad [\boldsymbol{\Pi}]
\end{aligned} \tag{6}
$$

Here $\boldsymbol{\psi} \in \mathbb{R}^N$, $\boldsymbol{\varphi} \in \mathbb{R}^{T^d}$, and $\boldsymbol{\beta} \in \mathbb{R}^{T^d \times k}$ are the dual variables, $\boldsymbol{\mu} = \frac{1}{T^d}\mathbf{1}_{T^d}$, $\boldsymbol{\nu} = \frac{1}{N}\mathbf{1}_N$ are the marginals and $\bar{\boldsymbol{X}} = \frac{1}{T^d N}\mathbf{1}_{T^d}\mathbf{1}_N^\top \boldsymbol{X}$ is the mean-independence constraint. The solution of these linear programs results in the convex potentials $\boldsymbol{\psi}(\boldsymbol{x}, \boldsymbol{y})$ and $\boldsymbol{\beta}(\boldsymbol{u})^\top \boldsymbol{x} + \boldsymbol{\varphi}(\boldsymbol{u})$, where the former is discretized over data samples and the latter over quantile levels. The estimated CVQF is then the transport map between the measures of the quantile levels and the data samples, and is given by $\widehat{Q}_{\mathbf{Y}|\mathbf{X}}(\boldsymbol{u}; \boldsymbol{x}) = \nabla_{\boldsymbol{u}}\left\{\boldsymbol{\beta}(\boldsymbol{u})^\top \boldsymbol{x} + \boldsymbol{\varphi}(\boldsymbol{u})\right\}$ (see also Appendix A.6). Notably, co-monotonicity of the CVQF arises due to it being the gradient of a convex function.

**Limitations of existing methods.** The primal VQR problem (5) has $T^d \cdot N$ variables and $T^d \cdot (k+1) + N$ constraints. Solving the above linear programs with general-purpose solvers has cubic complexity in the number of variables and constraints and is thus intractable even for modestly-sized VQR problems (see fig. A9). Numerous works proposed fast solvers for variants of OT problems. A common approach for scaling OT to large samples is to introduce an entropic regularization term $-\varepsilon \sum_{i,j} \Pi_{i,j} \log \Pi_{i,j}$ to the primal objective. As shown by Cuturi (2013), this regularized formulation allows using the Sinkhorn algorithm which provides significant speedup. Other solvers have been proposed for OT variants with application-specific regularization terms on $\boldsymbol{\Pi}$ (Ferradans et al., 2013; Solomon et al., 2015; Benamou et al., 2015). However, the addition of the mean-independence constraint on $\boldsymbol{\Pi}$ renders these solvers inapplicable for VQR. For example, the Sinkhorn algorithm would require three projections per iteration: one for each marginal, and one for the mean-independence constraint. Crucially, for the latter constraint the projection onto the feasible set has no closed-form solution and thus solving VQR via Sinkhorn is very slow in practice.

**Relaxed dual formulation.** Another approach for scaling OT is by solving its dual formulation. Genevay et al. (2016) showed that the relaxed dual version of the standard OT problem is amenable to stochastic optimization and can thus scale to large samples. The relaxed dual formulation of VQR is obtained from eq. (6) (see appendix A.4), and can we written as

$$
\min_{\boldsymbol{\psi},\boldsymbol{\beta}} \boldsymbol{\psi}^\top \boldsymbol{\nu} + \text{tr}\left(\boldsymbol{\beta}^\top \bar{\boldsymbol{X}}\right) + \varepsilon \sum_{i=1}^{T^d} \mu_i \log\left(\sum_{j=1}^{N} \exp\left(\frac{1}{\varepsilon}\left(\boldsymbol{u}_i^\top \boldsymbol{y}_j - \boldsymbol{\beta}_i^\top \boldsymbol{x}_j - \psi_j\right)\right)\right), \tag{7}
$$

where $\varepsilon$ controls the exactness of the the objective; as $\varepsilon$ decreases, the relaxed dual more closely approximates the exact dual. Of note are some important properties of this formulation: (i) it encodes the mean-independence constraint in the objective; (ii) it is equivalent to an entropic-regularized version of the VQR OT primal (5) (see appendix A.5); (iii) it is an unconstrained convex objective amenable to stochastic-gradient based optimization. The key drawbacks of this approach are the linear specification of the resulting CVQF in $\boldsymbol{x}$, and the potential violation of co-monotonicity due to relaxation of the constraints. We address the former limitation in Section 4 and the latter in Section 5.

**SGD-based solver.** The relaxed dual formulation of VQR (7) involves only $T^d \cdot k + N$ optimization variables and the objective is amenable to GPU-based acceleration, as it involves only dense-matrix

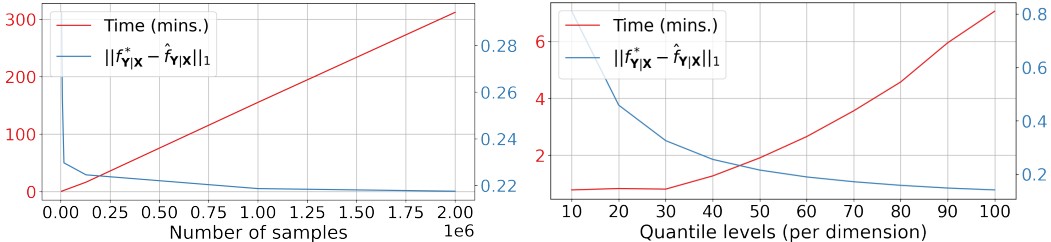

Figure 2: **Proposed VQR solver scales to large $N$ and $T$ with improving accuracy.** Computed on MVN dataset with $d = 2$ and $k = 10$. Blue curve shows KDE-L1 distance, defined in section 6. Left: Sweeping $N$; $T = 50$ and $B_N = 50k$. Right: Sweeping $T$; $N = 100k$, and $B_T = 2500$.

multiplications and pointwise operations. However, calculating this objective requires materializing two $T^d \times N$ inner-product matrices, causing the memory footprint to grow exponentially with $d$. This problem is exacerbated when $\mathbf{X}$ is high-dimensional, as it must also be kept in memory. These memory requirements severely limit the problem size which can be feasibly solved on general-purpose GPUs. Thus, naive stochastic gradient descent (SGD) with mini-batches of samples to (7) is insufficient for obtaining a scalable solver.

To attain a constant memory footprint w.r.t. $T$ and $N$, we sample data points from $\{(\boldsymbol{x}_j, \boldsymbol{y}_j)\}_{j=1}^{N}$ together with vector quantile levels from $\{\boldsymbol{u}_i\}_{i=1}^{T^d}$ and evaluate eq. (7) on these samples. Contrary to standard SGD, when sampling quantile levels, we select the corresponding entries from $\boldsymbol{\beta}$ and $\boldsymbol{\mu}$; likewise, when sampling data points, we select the corresponding entries from $\boldsymbol{\psi}$. Thus, by setting a fixed batch size for both data points and levels, we solve VQR with a constant memory footprint irrespective of problem size (fig. 2). Moreover, we observe that in practice, using SGD adds only negligible optimization error to the solution, with smaller batches producing more error as expected (Appendix fig. A10a). Refer to Appendix B for a convergence analysis of this approach and to Appendix G for implementation details.

## 4 NONLINEAR VQR

In this section we propose an extension to VQR which allows the estimated model to be non-linear in the covariates. Carlier et al. (2016) proved the existence and uniqueness of a CVQF which satisfies strong representation, i.e., $\mathbf{Y}|(\mathbf{X} = \boldsymbol{x}) = Q_{\mathbf{Y}|\mathbf{X}}(\mathbf{U}; \boldsymbol{x})$, and is co-monotonic w.r.t. $\mathbf{U}$. In order to estimate this function from finite samples, they further assumed a linear specification in both $\boldsymbol{u}$ and $\boldsymbol{x}$, given by the model $\widehat{Q}_{\mathbf{Y}|\mathbf{X}}^{L}(\boldsymbol{u}; \boldsymbol{x}) = \boldsymbol{B}(\boldsymbol{u})^{\top}\boldsymbol{x} + \boldsymbol{a}(\boldsymbol{u})$. This results in the OT problem with the mean-independence constraint (5). However, in practice and with real-world datasets, there is no reason to assume that such a specification is valid. In cases where the true CVQF is a non-linear function of $\boldsymbol{x}$ this model is mis-specified and the estimated CVQF does not satisfy strong representation.

**Extension to nonlinear specification.** We address the aforementioned limitation by modelling the CVQF as a non-linear function of the covariates parametrized by $\boldsymbol{\theta}$, i.e., $\widehat{Q}_{\mathbf{Y}|\mathbf{X}}^{NL}(\boldsymbol{u}; \boldsymbol{x}) = \boldsymbol{B}(\boldsymbol{u})^{\top}g_{\boldsymbol{\theta}}(\boldsymbol{x}) + \boldsymbol{a}(\boldsymbol{u})$. We fit $\boldsymbol{\theta}$ *jointly* with the regression coefficients $\boldsymbol{B}, \boldsymbol{a}$. The rationale is to parametrize an embedding $g_{\boldsymbol{\theta}}(\boldsymbol{x})$ such that the model is $Q_{\mathbf{Y}|\mathbf{X}}^{NL}$ is better specified than $Q_{\mathbf{Y}|\mathbf{X}}^{L}$ in the sense of strong representation. Encoding this nonlinear CVQF model into the exact formulations of VQR (eqs. (5) and (6)) will no longer result in a linear program. However, the proposed relaxed dual formulation of VQR (7) can naturally incorporate the aforementioned non-linear transformation of the covariates, as follows

$$\min_{\boldsymbol{\psi}, \boldsymbol{\beta}, \boldsymbol{\theta}} \ \boldsymbol{\psi}^{\top}\boldsymbol{\nu} + \mathrm{tr}\left(\boldsymbol{\beta}^{\top}\bar{\boldsymbol{G}}_{\boldsymbol{\theta}}(\boldsymbol{X})\right) + \varepsilon \sum_{i=1}^{T^d} \mu_i \log\left(\sum_{j=1}^{N} \exp\left(\frac{1}{\varepsilon}\left(\boldsymbol{u}_i^{\top}\boldsymbol{y}_j - \boldsymbol{\beta}_i^{\top}g_{\boldsymbol{\theta}}(\boldsymbol{x}_j) - \psi_j\right)\right)\right), \quad (8)$$

where $\bar{\boldsymbol{G}}_{\boldsymbol{\theta}}(\boldsymbol{X}) = \mathbf{1}_{T^d}\mathbf{1}_N^{\top}g_{\boldsymbol{\theta}}(\boldsymbol{X})/(T^d N)$ is the empirical mean after applying $g_{\boldsymbol{\theta}}$ to each sample. We optimize the above objective with the modified SGD approach described in section 3. The estimated non-linear CVQF can then be recovered by $\widehat{Q}_{\mathbf{Y}|\mathbf{X}}^{NL}(\boldsymbol{u}; \boldsymbol{x}) = \nabla_{\boldsymbol{u}}\left\{\boldsymbol{\beta}(\boldsymbol{u})^{\top}g_{\boldsymbol{\theta}}(\boldsymbol{x}) + \boldsymbol{\varphi}(\boldsymbol{u})\right\}$.

A crucial advantage of our nonlinear CVQF model is that $g_{\boldsymbol{\theta}}$ may embed $\boldsymbol{x} \in \mathbb{R}^k$ into a different dimension, $k' \neq k$. This means that VQR can be performed, e.g., on a lower-dimensional projection

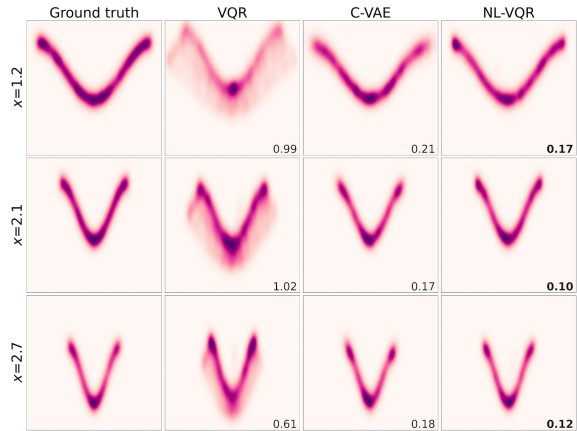

Figure 3: **NL-VQR quantitatively and qualitatively outperforms other methods in conditional distribution estimation.** Comparison of kernel-density estimates of samples drawn from VQR, CVAE, and NL-VQR, on the conditional-banana dataset. Models were trained with $N = 20k$ samples; for VQR $T = 50$ was used. Numbers depict the KDE-L1 metric. Lower is better.

of $\boldsymbol{x}$ or on a "lifted" higher-dimensional representation. For a low-dimensional $\mathbf{X}$ which has a highly-nonlinear relationship with the target $\mathbf{Y}$, lifting it into a higher dimension could result in the linear regression model (with coefficients $\boldsymbol{B}$ and $\boldsymbol{a}$) to be well-specified. Conversely, for high-dimensional $\mathbf{X}$ which has intrinsic low-dimensionality (e.g., an image), $g_{\boldsymbol{\theta}}$ may encode a projection operator which has inductive bias, and thus exploits the intrinsic structure of the feature space. This allows one to choose a $g_{\boldsymbol{\theta}}$ suitable for the problem at hand, for example, a convolutional neural network in case of an image or a graph neural network if $\boldsymbol{x}$ lives on a graph. In all our experiments, we indeed find that a learned non-linear feature transformation substantially improves the representation power of the estimated CVQF (see fig. 3; Appendix figs. A13 and A14).

**Simultaneous Quantile Regression (SQR).** SQR is the task of simultaneously resolving multiple conditional quantiles of a scalar variable, i.e. a possibly non-linear version of eq. (3). Nonlinear SQR has been well-studied, and multiple approaches have been proposed (Brando et al., 2022; Tagasovska & Lopez-Paz, 2019). We highlight that SQR can be performed as a special case of NL-VQR (8) when $d = 1$. See Appendix fig. A12 for an example. Other SQR approaches enforce monotonicity of the CQF via different ad-hoc techniques, while with NL-VQR the monotonicity naturally emerges from the OT formulation as explained in section 3; thus it is arguably a preferable approach. To our knowledge, OT-based SQR has not been demonstrated before.

**Prior attempts at nonlinear high-dimensional quantiles.** Feldman et al. (2021) proposed an approach for constructing high-dimensional confidence regions, based on a *conditional variational autoencoder* (CVAE). A crucial limitation of this work is that it does not infer the CVQF; their resulting confidence regions are therefore not guaranteed to be quantile contours (see fig. 1a), because they do not satisfy the defining properties of vector quantiles (eqs. (1) and (2)).

## 5 VECTOR MONOTONE REARRANGEMENT

Solving the relaxed dual(7) may lead to violation of co-monotonicity in $Q_{\mathbf{Y}|\mathbf{X}}$, since the exact constraints in eq. (6) are not enforced. This is analogous to the quantile crossing problem in the scalar case, which can also manifest when QR is performed separately for each quantile level (Chernozhukov et al., 2010) (Appendix fig. A12d). In what follows, we propose a way to overcome this limitation.

Consider the case of scalar quantiles, i.e., $d = 1$. Denote $\widehat{Q}_{\mathrm{Y}|\mathbf{X}}(u; \boldsymbol{x})$ as an estimated CQF, which may be non-monotonic in $u$ due to estimation error and thus may not be a valid CQF. One may convert $\widehat{Q}_{\mathrm{Y}|\mathbf{X}}(u; \boldsymbol{x})$ into a monotonic $\widetilde{Q}_{\mathrm{Y}|\mathbf{X}}(u; \boldsymbol{x})$ through rearrangement as follows. Consider a random variable defined as $\widehat{\mathrm{Y}}|\mathbf{X} := \widehat{Q}_{\mathrm{Y}|\mathbf{X}}(\mathrm{U}; \boldsymbol{x})$ where $\mathrm{U} \sim \mathbb{U}[0, 1]$. Its CDF and inverse CDF are given by $F_{\widehat{\mathrm{Y}}|\mathbf{X}}(y; \boldsymbol{x}) = \int_0^1 \mathbb{I}\left\{\widehat{Q}_{\mathrm{Y}|\mathbf{X}}(u; \boldsymbol{x}) \leq y\right\} du$, and $Q_{\widehat{\mathrm{Y}}|\mathbf{X}}(u; \boldsymbol{x}) = \inf\left\{y : F_{\widehat{\mathrm{Y}}|\mathbf{X}}(y; \boldsymbol{x}) \geq u\right\}$. $Q_{\widehat{\mathrm{Y}}|\mathbf{X}}(u; \boldsymbol{x})$ is the true CQF of $\widehat{\mathrm{Y}}|\mathbf{X}$ and is thus necessarily monotonic. It can be shown that $Q_{\widehat{\mathrm{Y}}|\mathbf{X}}(u; \boldsymbol{x})$ is no worse an estimator for the true CQF $Q_{\mathrm{Y}|\mathbf{X}}(u; \boldsymbol{x})$ than $\widehat{Q}_{\mathrm{Y}|\mathbf{X}}(u; \boldsymbol{x})$ in the $L_p$-norm sense (Chernozhukov et al., 2010). In practice, this rearrangement is performed in the scalar case by sorting the discrete estimated CQF. Rearrangement has no effect if $\widehat{Q}_{\mathrm{Y}|\mathbf{X}}(u; \boldsymbol{x})$ is already monotonic.

Here, we extend the notion of rearrangement to the case of $d > 1$. As before, define $\widehat{\mathbf{Y}}|\mathbf{X} := \widehat{Q}_{\mathbf{Y}|\mathbf{X}}(\mathbf{U}; \boldsymbol{x})$ where $\mathbf{U} \sim \mathbb{U}[0, 1]^d$, and $\widehat{Q}_{\mathbf{Y}|\mathbf{X}}(u; \boldsymbol{x})$ is the estimated CVQF. If it is not co-monotonic,

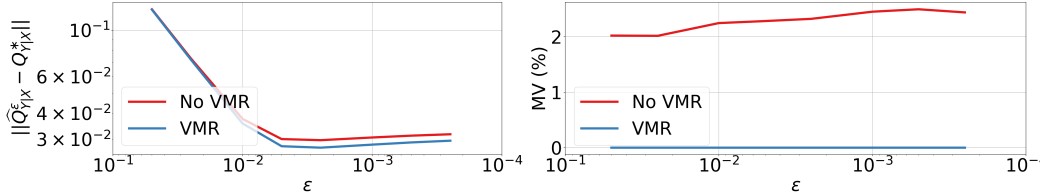

Figure 4: **VMR improves strong representation (QFD; left), and completely eliminates mono-tonicity violations (MV; right).** VMR allows for smaller $\varepsilon$ values, thus improving accuracy (lower QFD) without compromising monotonicity (zero MV). Without VMR, MV increases (right, red) as the problem becomes more exact (smaller $\varepsilon$). MVN dataset; $N = 20k$, $T = 50$, $d = 2$, $k = 1$.

as defined by eq. (1), we can compute a co-monotonic $Q_{\widehat{\mathbf{Y}}|\mathbf{X}}(u; \boldsymbol{x})$ by calculating the vector quantiles of $\widehat{\mathbf{Y}}|\mathbf{X} = \boldsymbol{x}$, separately for each $\boldsymbol{x}$. We emphasize that this amounts to solving the simpler vector quantile *estimation* problem for a specific $\boldsymbol{x}$, as opposed to the vector quantile *regression* problem.

Let $\widehat{\boldsymbol{Q}} = [\widehat{\boldsymbol{q}}_j]_j \in \mathbb{R}^{T^d \times d}$ be the estimated CVQF $\widehat{Q}_{\mathbf{Y}|\mathbf{X}}(\boldsymbol{u}; \boldsymbol{x})$ sampled at $T^d$ levels $\boldsymbol{U} = [\widehat{\boldsymbol{u}}_i]_i \in \mathbb{R}^{T^d \times d}$ by solving VQR. To obtain $Q_{\widehat{\mathbf{Y}}|\mathbf{X}}(u; \boldsymbol{x})$ sampled at $\boldsymbol{U}$ we solve

$$\max_{\pi_{i,j} \geq 0} \sum_{i,j=1}^{T^d} \pi_{i,j} \boldsymbol{u}_i^\top \widehat{\boldsymbol{q}}_j \quad \text{s.t.} \quad \boldsymbol{\Pi}^\top \mathbf{1} = \boldsymbol{\Pi} \mathbf{1} = \frac{1}{T^d} \mathbf{1}, \tag{9}$$

and then compute $\widetilde{\boldsymbol{Q}} = T^d \cdot \boldsymbol{\Pi}\widehat{\boldsymbol{Q}}$. We define this procedure as *vector monotone rearrangement* (VMR). The existence and uniqueness of a co-monotonic function, mapping between the measures of $\mathbf{U}$ and $\widehat{Q}_{\mathbf{Y}|\mathbf{X}}(\mathbf{U}; \boldsymbol{x})$ is due to the Brenier's theorem as explained in section 1 (Brenier, 1991; McCann, 1995; Villani, 2021). VMR can be interpreted as "vector sorting" of the discrete CVQF estimated with VQR, since it effectively permutes the entries of $\widehat{\boldsymbol{Q}}$ such that the resulting $\widetilde{\boldsymbol{Q}}$ is co-monotonic with $\boldsymbol{U}$. Note that eq. (9) is an OT problem with the inner-product matrix as the ground cost, and simple constraints on the marginals of the transport plan $\boldsymbol{\Pi}$. Thus, VMR (9) is significantly simpler to solve *exactly* than VQR (5). We leverage fast, off-the-shelf OT solvers from the POT library (Flamary et al., 2021; Bonneel et al., 2011) and apply VMR as a post-processing step, performed after the estimated CVQF $\widehat{Q}_{\mathbf{Y}|\mathbf{X}}(u; \boldsymbol{x})$ is evaluated for a specific $\boldsymbol{x}$.

In the 1d case, quantile crossings before and after VMR correction can be readily visualized (Appendix figs. A12d and A12e). For $d > 1$, monotonicity violations manifest as $(\boldsymbol{u}_i - \boldsymbol{u}_j)^\top (Q(\boldsymbol{u}_i) - Q(\boldsymbol{u}_j)) < 0$ (Appendix fig. A11). Figure 4 demonstrates that co-monotonicity violations are completely eliminated and that strong representation strictly improves by applying VMR.

## 6 EXPERIMENTS

We use four synthetic and four real datasets which are detailed in Appendices D.1 and D.2. Except for the MVN dataset, which was used for the scale and optimization experiments, the remaining three synthetic datasets were carefully selected to be challenging since they exhibit complex nonlinear relationships between $\mathbf{X}$ and $\mathbf{Y}$ (see e.g. fig. 1b). We evaluate using the following metrics (detailed in Appendix E): (i) KDE-L1, an estimate of distance between distributions; (ii) QFD, a distance measured between an estimated CVQF and its ground truth; (iii) Inverse CVQF entropy; (iv) Monotonicity violations; (v) Marginal coverage; (vi) Size of $\alpha$-confidence set. The first three metrics serve as a measure of strong representation (2), and can only be used for the synthetic experiments since they require access to the data generating process. The last two metrics are used for real data experiments, as they only require samples from the joint distribution. Implementation details for all experiments can be found in Appendix G.

### 6.1 SCALE AND OPTIMIZATION

To demonstrate the scalability of our VQR solver and its applicability to large problems, we solved VQR under multiple increasing values of $N$ and $T$, while keeping all other data and training settings fixed, and measured both wall time and KDE-L1. We used up to $N = 2 \cdot 10^6$ data points, $d = 2$ dimensions, and $T = 100$ levels per dimension ($T^d = 100^2$ levels in total), while sampling both data points and quantile levels stochastically, thus keeping the memory requirement fixed throughout. This

enabled us to run the experiment on a commodity 2080Ti GPU with 11GB of memory. Optimization experiments, showing the effects of $\varepsilon$ and batch sizes on convergence, are presented in Appendix F.

Figure 2 presents these results. Runtime increases linearly with $N$ and quadratically with $T$, as can be expected for $d = 2$. KDE-L1 consistently improves when increasing $N$ and $T$, showcasing improved accuracy in distribution estimation, especially as more quantile levels are estimated. To the best of our knowledge, this is the first time that large-scale VQR has been demonstrated.

## 6.2 SYNTHETIC DATA EXPERIMENTS

Here our goal is to evaluate the estimation error (w.r.t the ground-truth CVQF) and sampling quality (when sampling from the CVQF) of nonlinear VQR. We use the conditional-banana, rotating stars and synthetic glasses datasets, where the assumption of a linear CVQF is violated.

**Baselines.** We use both linear VQR and *conditional variational autoencoders* (CVAE) (Feldman et al., 2021) as strong baselines for estimating the conditional distribution of $\mathbf{Y}|\mathbf{X}$. We emphasize that CVAE only allows sampling from the estimated conditional distribution; it does not estimate quantiles, while VQR allows both. Thus, we could compare VQR with CVAE only on the KDE-L1 metric, which is computed on samples. To the best of our knowledge, besides VQR there is no other generative model capable of estimating CVQFs.

**Conditional Banana.** The results of the conditional-banana experiment, comparing VQR, NL-VQR and CVAE, are presented in fig. 3, Appendix table A2 and fig. A13. Two-dimensional KDEs of conditional distributions for three values of $x$ are depicted per method (fig. 3). Qualitatively, sampling from either NL-VQR or CVAE produces accurate estimations of the ground truth distribution, when compared to linear VQR. This indicates that the linear VQR model is mis-specified for this data, which is further corroborated by the entropy metric (table A2). The entropy of the inverse CVQF is lower for linear VQR, indicating non-uniformity and therefore mis-specification. Quantitatively in terms of the KDE-L1 metric, NL-VQR outperforms CVAE by 23%, and in terms of QFD, nonlinear VQR results in 70% lower error than linear VQR.

**Rotating Stars.** The results of the rotating stars experiment with the same baselines as above, are presented in Appendix fig. A14 and table A3. As illustrated in fig. 1b, this dataset is highly challenging due to the non-trivial dependence between $\mathbf{X}$ and the tails of $\mathbf{Y}|\mathbf{X}$. We observe that NL-VQR qualitatively and quantitatively outperforms both CVAE and linear VQR by a large margin. It is therefore the only evaluated method that was able to faithfully model the data distribution.

These results highlight that NL-VQR is significantly better at estimating CVQFs compared to VQR. Another point of note is that the CVAE model required 35 minutes to train, while VQR and NL-VQR were trained within less than 4 minutes. NL-VQR therefore outperforms CVAE with almost an order of magnitude speedup in training time, and outperforms VQR while requiring similar runtime. Additional synthetic experiments, showcasing OT-based SQR and VMR are presented in Appendix F.

## 6.3 REAL DATA EXPERIMENTS

VQR has numerous potential applications, as detailed in section 1. Here we showcase one immediate and highly useful application, namely distribution-free uncertainty estimation for vector-valued targets. Given $P_{(\mathbf{X},\mathbf{Y})}$ and a confidence level $\alpha \in (0,1)$, the goal is to construct a conditional $\alpha$-confidence set $\mathcal{C}_\alpha(\boldsymbol{x})$ such that it has marginal coverage of $1 - \alpha$, defined as $\mathbb{P}\left[\mathbf{Y} \in \mathcal{C}_\alpha(\mathbf{X})\right] = 1 - \alpha$. A key requirement from an uncertainty estimation method is to produce a small $\mathcal{C}_\alpha(\boldsymbol{x})$ which satisfies marginal coverage, without any distributional assumptions (Romano et al., 2019).

**Baselines.** We compare nonlinear VQR against (i) Separable linear QR (Sep-QR); (ii) Separable nonlinear QR (Sep-NLQR); (iii) linear VQR. For Sep-QR and Sep-NLQR the estimated CVQF is

$$\widehat{Q}^{\text{Sep}}_{\mathbf{Y}|\mathbf{X}}(\boldsymbol{u}; \boldsymbol{x}) = \left[\widehat{Q}^{QR}_{Y_1|\mathbf{X}}(u_1; \boldsymbol{x}), \dots \widehat{Q}^{QR}_{Y_d|\mathbf{X}}(u_d; \boldsymbol{x})\right]^\top, \tag{10}$$

where $\widehat{Q}^{QR}_{Y_i|\mathbf{X}}(u_i; \boldsymbol{x})$ is obtained via 1d linear or nonlinear quantile regression of the variable $Y_i$. These separable baselines represent the basic approaches for *distribution-free* uncertainty estimation with vector-valued variables. They work well in practice and the size of the $\alpha$-confidence sets they produce serves an upper bound, due to their inherent independence assumption.

**Evaluation procedure.** The key idea for comparing distribution-free uncertainty estimation approaches is as follows. First, a nominal coverage rate $1 - \alpha^*$ is chosen. An $\alpha$-confidence set is then

constructed for each estimation method, such that it obtains the nominal coverage (in expectation). This is similar to calibration in conformal prediction (Sesia & Romano, 2021) since $\alpha$ is calibrated to control the size of the $\alpha$-confidence set. Finally, the size of the $\alpha$-confidence set is measured as the evaluation criterion. Appendix G contains experimental details and calibrated $\alpha$ values.

**Results.** Across the four datasets, NL-VQR acheives 34-60% smaller $\alpha$-confidence set size compared to the second-best performing method (section 6.3 and fig. A15; table A4). The reason for the superior performance of NL-VQR is its ability to accurately capture the shape of the conditional distribution, leading to small confidence sets with the same coverage (section 6.3 and fig. A16).

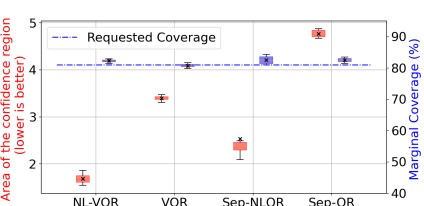

## 7 SOFTWARE

We provide our VQR solvers as part of a robust, well-tested python package, `vqr` (available in the supplementary materials). Our package implements fast solvers with support for GPU-acceleration and has a familiar `sklearn`-style API. It supports any number of dimensions for both input features ($k$) and target variables ($d$), and allows for arbitrary neural networks to be used as the learned non-linear feature transformations, $g_\theta$. The package provides tools for estimating vector quantiles, vector ranks, vector quantile contours, and performing VMR as a refinement step after fitting VQR. To the best of our knowledge, this would be the first publicly available tool for estimating conditional vector quantile functions at scale. See Appendix H for further details.

## 8 CONCLUSION

In this work, we proposed NL-VQR, and a scalable approach for performing VQR in general. NL-VQR overcomes a key limitation of VQR, namely the assumption that the CVQF is linear in $\mathbf{X}$. Our approach allows modelling conditional quantiles by embedding $\mathbf{X}$ into a space where VQR is better specified and further to exploit the

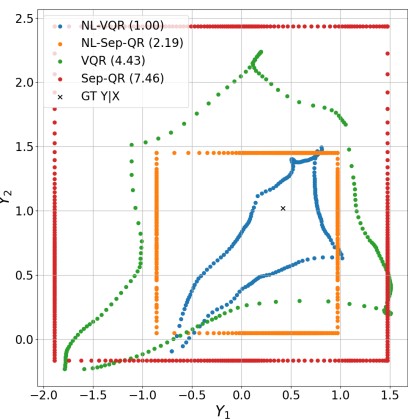

Figure 5: **NL-VQR produces substantially smaller confidence sets than other methods at a given confidence level.** Top: Mean $\mathcal{C}_\alpha$ size; Bottom: Example $\mathcal{C}_\alpha(\boldsymbol{x})$ for a specific $\boldsymbol{x}$. Calculated on house prices test-set. Full details in figs. A15 and A16.

structure of the domain of $\mathbf{X}$ (via a domain-specific models like CNNs). We proposed a new relaxed dual formulation and custom solver for the VQR problem and demonstrated that we can perform VQR with millions of samples. As far as we know, large scale estimation of CVQFs or even VQFs has not been previously shown. Moreover, we resolved the issue of high-dimensional quantile crossings by proposing VMR, a refinement step for estimated CVQFs. We demonstrated, through exhaustive synthetic and real data experiments, that NL-VQR with VMR is by far the most accurate way to model CVQFs. Finally, based on the real data results, we argue that NL-VQR should be the primary approach for distribution-free uncertainty estimation, instead of separable approaches which assume independence.

**Limitations.** As with any multivariate density estimation task which does not make distributional assumptions, our approach suffers from the curse of dimensionality, especially in the target dimension. Overcoming this limitation requires future work, and might entail, e.g., exploiting the structure of the domain of $\mathbf{Y}$. This could be achieved by leveraging recent advances in high-dimensional neural OT (Korotin et al., 2021). Another potential limitation is that the nonlinear transformation $g_\theta(\boldsymbol{x})$ is shared across quantile levels (it is not a function of $\boldsymbol{u}$), though evaluating whether this is truly a limiting assumption in practice requires further investigation.

In conclusion, although quantile regression is a very popular tool, vector quantile regression is arguably far less known, accessible, and usable in practice due to lack of adequate tools. This is despite that fact that it is a natural extension of QR, which can be used for general statistical inference tasks. We present the community with an off-the-shelf method for performing VQR in the real world. We believe this will contribute to many existing applications, and inspire a wide-range of new applications, for which it is currently prohibitive or impossible to estimate CVQFs.

ACKNOWLEDGEMENTS

Y.R. was supported by the Israel Science Foundation (grant No. 729/21). Y.R. thanks Shai Feldman and Stephen Bates for discussions about vector quantile regression, and the Career Advancement Fellowship, Technion, for providing research support. A.A.R., S.V., and A.M.B. were partially supported by the European Research Council (ERC) under the European Unions Horizon 2020 research and innovation programme (grant agreement No. 863839), by the Council For Higher Education - Planning & Budgeting Committee, and by the Israeli Smart Transportation Research Center (ISTRC).

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

## A DERIVATIONS

In this section, we present the following derivations:

- Formulating one-dimensional QR as a correlation-maximization problem (Appendix A.1).

- Rephrasing correlation maximization as one-dimensional optimal transport (Appendix A.2).

- Extension of the OT-based formulation for QR to the multi-dimensional targets (Appendix A.3).

- Relaxing the dual formulation of the OT-based VQR problem (Appendix A.4).

- The equivalence between the entropic-regularized version of the OT-based primal and the relaxed dual formulation (Appendix A.5).

- Calculating the conditional (vector) quantile functions from the dual variables (Appendix A.6).

The derivations in this section are not rigorous mathematical proofs; instead they are meant to show an easy-to-follow way to obtain the high-dimensional VQR relaxed dual objective (which we eventually solve), by starting from the well-known one-dimensional QR based on the pinball loss.

### A.1 QUANTILE REGRESSION AS CORRELATION MAXIMIZATION

The pinball loss defined above can be written as,

$$
\rho_u(z) = \begin{cases} uz, & z > 0 \\ (u-1)z, & z \leq 0 \end{cases}
$$
$$
= \begin{cases} uz - z + z, & z > 0 \\ (u-1)z, & z \leq 0 \end{cases}
$$
$$
= z^+ + (u-1)z
$$

where $z^+ \triangleq \max\{0, z\}$. Note that we also define $z^- \triangleq \max\{0, -z\}$ which we use later. Given the continuous joint distribution $P_{\mathbf{X},\mathrm{Y}}$, and assuming a linear model relates the $u$-th quantile of Y to $\mathbf{X}$, then performing quantile regression involves minimizing

$$
\min_{a_u, \boldsymbol{b}_u} \mathbb{E}_{\mathbf{X},\mathrm{Y}} \left[ \left( \mathrm{Y} - \boldsymbol{b}_u^\top \mathbf{X} - a_u \right)^+ - (1-u) \left( \mathrm{Y} - \boldsymbol{b}_u^\top \mathbf{X} - a_u \right) \right]. \tag{11}
$$

Define $Z(a_u, \boldsymbol{b}_u) \triangleq \mathrm{Y} - \boldsymbol{b}_u^\top \mathbf{X} - a_u$, the above problem can be written as

$$
\min_{a_u, \boldsymbol{b}_u} \mathbb{E}_{\mathbf{X},\mathrm{Y}} \left[ Z(a_u, \boldsymbol{b}_u)^+ - (1-u) Z(a_u, \boldsymbol{b}_u) \right]. \tag{12}
$$

Define $P_u \triangleq Z(a_u, \boldsymbol{b}_u)^+$ and $N_u \triangleq Z(a_u, \boldsymbol{b}_u)^-$ as the positive and negative deviations from the true $u$-th quantile, then notice that (i) $P_u \geq 0$ and $N_u \geq 0$; (ii) $P_u - N_u = Z(a_u, \boldsymbol{b}_u)$. Introducing $P_u$ and $N_u$ as slack variables, we can rewrite the above optimization problem as

$$
\min_{a_u, \boldsymbol{b}_u, P_u, N_u} \mathbb{E}_{\mathbf{X},\mathrm{Y}} \left[ P_u - (1-u) Z(a_u, \boldsymbol{b}_u) \right]
$$
$$
\text{s.t.,}
$$
$$
P_u \geq 0, N_u \geq 0
$$
$$
P_u - N_u = Z(a_u, \boldsymbol{b}_u), \text{ with multiplier } [V_u].
$$

Solving the above problem is equivalent to solving the Lagrangian formulation

$$
\min_{P_u, N_u, a_u, \boldsymbol{b}_u} \max_{V_u} \mathbb{E}_{\mathbf{X},\mathrm{Y}} \left[ P_u - (1-u) Z(a_u, \boldsymbol{b}_u) - V_u \left( P_u - N_u - Z(a_u, \boldsymbol{b}_u) \right) \right]
$$
$$
\text{s.t.,}
$$
$$
P_u \geq 0, N_u \geq 0.
$$

Substituting $Z_u(a_u, \boldsymbol{b}_u) = Y - \boldsymbol{b}_u^\top \mathbf{X} - a$, we get

$$\min_{P_u, N_u, a_u, \boldsymbol{b}_u} \max_{V_u} \mathbb{E}_{\mathbf{X}, Y} \left[ P_u - (1-u) \left( Y - \boldsymbol{b}_u^\top \mathbf{X} - a_u \right) - V_u \left( P_u - N_u - Y + \boldsymbol{b}_u^\top \mathbf{X} + a_u \right) \right]$$
$$\text{s.t.,}$$
$$P_u \geq 0, N_u \geq 0.$$

In the first term, Y can be omitted because it is independent of the optimization variables, and in the second term, Y can be separated, this yields

$$\max_{V_u} \mathbb{E}_{\mathbf{X}, Y} \left[ V_u Y \right] + \min_{P_u, N_u, a_u, \boldsymbol{b}_u} \mathbb{E}_{\mathbf{X}, Y} \left[ P_u - (u-1) \left( \boldsymbol{b}_u^\top \mathbf{X} + a_u \right) - V_u \left( P_u - N_u + \boldsymbol{b}_u^\top \mathbf{X} + a_u \right) \right]$$
$$\text{s.t.,}$$
$$P_u \geq 0, N_u \geq 0.$$

Rewriting the terms by separating $a_u, \boldsymbol{b}_u, P_u, N_u$ yields

$$\max_{V_u} \mathbb{E}_{\mathbf{X}, Y} \left[ V_u Y \right] + \min_{P_u, N_u, a_u, \boldsymbol{b}_u} \mathbb{E}_{\mathbf{X}, Y} \left[ P_u (1 - V_u) + N_u V_u - a_u (V_u - (1-u)) - \boldsymbol{b}_u^\top \left( V_u \mathbf{X} - (1-u) \mathbf{X} \right) \right]$$
$$\text{s.t.,}$$
$$P_u \geq 0, N_u \geq 0.$$

By treating $P_u, N_u, a_u, \boldsymbol{b}_u$ as Lagrange multipliers we can obtain the dual formulation. Since $P_u, N_u \geq 0$, they must participate in inequality constraints, and since $a_u, \boldsymbol{b}_u$ are unconstrained, they participate in equality constraints. Thus,

$$\max_{V_u} \mathbb{E}_{\mathbf{X}, Y} \left[ V_u Y_u \right]$$
$$\begin{aligned}
\text{s.t. } & V_u \geq 0 & [N_u] \\
& V_u \leq 1 & [P_u] \\
& \mathbb{E}_{\mathbf{X}, Y} \left[ V_u \right] = (1 - u) & [a_u] \\
& \mathbb{E}_{\mathbf{X}, Y} \left[ V_u \mathbf{X} \right] = (1 - u) \mathbb{E} \left[ \mathbf{X} \right] & [\boldsymbol{b}_u].
\end{aligned}$$

Note that (i) complementary slackness dictates that we always have $P_u, N_u \geq 0$, and that (ii) the last two constraints can be seen together as implying the mean independence condition of $\mathbb{E} \left[ \mathbf{X} | V \right] = \mathbb{E} \left[ \mathbf{X} \right]$.

Solving for multiple quantiles simultaneously can be achieved by minimizing the sum of above optimization problem $\forall u \in [0, 1]$. In addition we demand monotonicity of the estimated quantiles, such that $u \geq u' \implies \boldsymbol{b}_u^\top \mathbf{X} + a_u \geq \boldsymbol{b}_{u'}^\top \mathbf{X} + a_{u'}$. According to the KKT conditions, the solution to the above problem must satisfy complementary slackness, which in this case means

$$(1 - V_u) P_u = 0$$
$$V_u N_u = 0.$$

Following the definitions of $P_u$, $N_u$, we have the following relation between the estimate quantile $\boldsymbol{b}_u^\top \mathbf{X} + a_u$ and $V_u$:

$$\begin{cases}
Y > \boldsymbol{b}_u^\top \mathbf{X} + a_u \implies P_u > 0 & \implies V_u = 1 \\
Y < \boldsymbol{b}_u^\top \mathbf{X} + a_u \implies N_u > 0 & \implies V_u = 0 \\
Y = \boldsymbol{b}_u^\top \mathbf{X} + a_u \implies P_u = N_u = 0 & \implies 0 \leq V_u \leq 1
\end{cases}$$

Since $P_{\mathbf{X}, Y}$ is a continuous distribution, then for any $a_u, \boldsymbol{b}_u$, we have $\mathbb{P} \left[ Y = \boldsymbol{b}_u^\top \mathbf{X} + a_u \right] = 0$. Thus in practice we can ignore this case and write

$$V_u = \mathbb{I} \left\{ Y \geq \boldsymbol{b}_u^\top \mathbf{X} + a_u \right\}. \tag{13}$$

Therefore the monotonicity constraint translates to $V_u$ as

$$u \geq u' \implies \boldsymbol{b}_u^\top \mathbf{X} + a_u \geq \boldsymbol{b}_{u'}^\top \mathbf{X} + a_{u'} \iff V_u \leq V_{u'}.$$

Adding the monotonicity constraint on $V_u$ to the gives rise to a new dual problem:

$$
\max_{V_u} \int_0^1 \mathbb{E}_{\mathbf{X},\mathbf{Y}}\left[V_u Y_u\right] du
$$

$$
\begin{aligned}
\text{s.t. } & V_u \geq 0 && [N_u] \\
& V_u \leq 1 && [P_u] \\
& \mathbb{E}_{\mathbf{X},\mathbf{Y}}\left[V_u\right] = (1-u) && [a_u] \\
& \mathbb{E}_{\mathbf{X},\mathbf{Y}}\left[V_u \mathbf{X}\right] = (1-u)\mathbb{E}\left[\mathbf{X}\right] && [\boldsymbol{b}_u] \\
& u \geq u' \implies V_u \leq V_{u'}.
\end{aligned}
$$

Let us now consider a dataset $\{(\boldsymbol{x}_i, y_i)\}_{i=1}^N$ of i.i.d. samples from $P_{\mathbf{X},\mathbf{Y}}$ where $\boldsymbol{x}_i \in \mathbb{R}^k$ and $y_i \in \mathbb{R}$. We are interested in calculating $T$ quantiles for levels $u_1 = 0 < u_2 < \cdots < u_T \leq 1$. Furthermore, denote the sample mean $\bar{\boldsymbol{x}} \in \mathbb{R}^k$ where $\bar{x}_k = \sum_{i=1}^N x_{i,k}/N$. By discretizing the above problem, we arrive at

$$
\max_{V_{\tau,i}} \sum_{\tau=1}^T \sum_{i=1}^N V_{\tau,i} y_i
$$

$$
\begin{aligned}
\text{s.t. } & V_{\tau,i} \geq 0 && [N_{\tau,i}] \\
& V_{\tau,i} \leq 1 && [P_{\tau,i}] \\
& \frac{1}{N}\sum_{i=1}^N V_{\tau,i} = (1 - u_\tau) && [a_\tau] \\
& \frac{1}{N}\sum_{i=1}^N V_{\tau,i} x_{i,k} = (1 - u_\tau)\bar{x}_k && [b_{\tau,k}] \\
& V_{T,i} \leq V_{T-1,i}, \leq \ldots V_{1,i}.
\end{aligned}
$$

Now we can vectorize the above formulation by defining the matrix $\boldsymbol{V} \in \mathbb{R}^{T \times N}$ containing elements $V_{\tau,i}$, the first-order finite differences matrix

$$
\boldsymbol{D} = \begin{pmatrix}
1 & 0 & \cdots & & 0 \\
-1 & 1 & & & 0 \\
0 & -1 & & & \vdots \\
0 & 0 & & & \\
\vdots & \vdots & & 1 & 0 \\
0 & \cdots & & -1 & 1
\end{pmatrix} \in \mathbb{R}^{T \times T},
$$

and the vector of desired quantile levels, $\boldsymbol{u} = [u_1, \ldots, u_T]^\top$. The monotonicity constraint therefore becomes $\boldsymbol{v}_i^\top \boldsymbol{D} \geq 0 \ \forall i = 1, \ldots, N$ where $\boldsymbol{v}_i$ is the $i$-th column of $\boldsymbol{V}$. We also denote the covariates matrix $\boldsymbol{X} \in \mathbb{R}^{N \times k}$ and response vector $\boldsymbol{y} \in \mathbb{R}^N$. Thus, the problem vectorizes as,

$$
\max_{\boldsymbol{V}} \mathbf{1}_T^\top \boldsymbol{V} \boldsymbol{y}
$$

$$
\begin{aligned}
\text{s.t. } & V_{\tau,i} \geq 0 && [N_{\tau,i}] \\
& V_{\tau,i} \leq 1 && [P_{\tau,i}] \\
& \frac{1}{N}\boldsymbol{V}\mathbf{1}_N = (\mathbf{1}_T - \boldsymbol{u}) && [\boldsymbol{a}] \\
& \frac{1}{N}\boldsymbol{V}\boldsymbol{X} = (\mathbf{1}_T - \boldsymbol{u})\bar{\boldsymbol{x}}^\top && [\boldsymbol{B}] \\
& \boldsymbol{V}^\top \boldsymbol{D} \geq 0.
\end{aligned}
$$

where $\boldsymbol{a} \in R^T$ and $\boldsymbol{B} \in \mathbb{R}^{T \times k}$ are the dual variables which contain the regression coefficients per quantile level. We can observe the following for the above problem:

1. For any random variable, its zeroth quantile is smaller than or equal to any values the variable takes. In particular, for $\tau = 0$ we have $V_{0,i} = 1 \; \forall i$ because from eq. (13) we know that $V_{0,i}$ is the indicator that $y_i$ is greater that the zeroth quantile of $Y$.

2. The last constraint $\boldsymbol{V}^\top \boldsymbol{D} \geq 0$, which enforces non-increasing monotonicity along the quantile level, i.e., $V_{\tau,i} \leq V_{\tau',i} \; \forall \tau \geq \tau'$, also enforces that $V_{T,i} \geq 0$.

Therefore, we can observe that the first ($V_{\tau,i} \geq 0$), second ($V_{\tau,i} \leq 1$) and last ($\boldsymbol{V}^\top \boldsymbol{D} \geq 0$) constraints are partially-redundant and can be condensed into only two vectorized constraints: (1) $\boldsymbol{V}^\top \boldsymbol{D} \geq 0$ which ensures monotonicity and non-negativity of all elements in $\boldsymbol{V}$; (2) $\boldsymbol{V}^\top \boldsymbol{D} \boldsymbol{1}_T \leq \boldsymbol{1}_N$ which enforces that $V_{0,i} \leq 1 \; \forall i$. Notice that condensing the constraints in this manner comes with the inability to interpret the meaning of the Lagrange multipliers $P_{\tau,i}, N_{\tau,i}$. However, the advantage is that we have less constraints in total and the interpretability of the multipliers $\boldsymbol{a}$, $\boldsymbol{B}$ is maintained. Thus, we arrive at the following vectorized problem,

$$\max_{\boldsymbol{V}} \mathbf{1}_T{}^\top \boldsymbol{V} \boldsymbol{y}$$

$$\text{s.t.} \quad \frac{1}{N} \boldsymbol{V} \mathbf{1}_N = (\mathbf{1}_T - \boldsymbol{u}) \qquad\qquad [\boldsymbol{a}]$$

$$\frac{1}{N} \boldsymbol{V} \boldsymbol{X} = (\mathbf{1}_T - \boldsymbol{u})\bar{\boldsymbol{x}}^\top \qquad\qquad [\boldsymbol{B}]$$

$$\boldsymbol{V}^\top \boldsymbol{D} \geq 0$$

$$\boldsymbol{V}^\top \boldsymbol{D} \mathbf{1}_T \leq \mathbf{1}_N.$$

## A.2 Correlation Maximization as Optimal Transport

Following Carlier et al. (2016), the above problem can be re-formulated as an Optimal Transport problem, i.e. the problem of finding a mapping between two probability distributions.

Assume we are now interested in estimating the quantiles of Y at $T$ uniformly-sampled levels, $[u_1, \ldots, u_T]^\top = \left[\frac{1}{T}, \frac{2}{T}, \ldots, \frac{T}{T}\right]^\top$. In the above problem, one can decompose the objective as

$$\mathbf{1}_T{}^\top \boldsymbol{V} \boldsymbol{y} = \left(\mathbf{1}_T{}^\top \boldsymbol{D}^{-\top}\right)\left(\boldsymbol{D}^\top \boldsymbol{V}\right)\boldsymbol{y} = \left(\boldsymbol{D}^{-1}\mathbf{1}_T\right)^\top \left(\boldsymbol{D}^\top \boldsymbol{V}\right)\boldsymbol{y}.$$

If we then denote

$$\boldsymbol{\Pi} = \frac{1}{N}\boldsymbol{D}^\top \boldsymbol{V} \in \mathbb{R}^{T \times N}$$

$$\boldsymbol{u} = \frac{1}{T}\boldsymbol{D}^{-1}\mathbf{1}_T = \left[\frac{1}{T}, \frac{2}{T}, \ldots, \frac{T}{T}\right]^\top \in \mathbb{R}^T,$$

then we can write the objective as $NT \cdot \boldsymbol{u}^\top \boldsymbol{\Pi} \boldsymbol{y}$. In addition, denote

$$\boldsymbol{\mu} = \boldsymbol{D}^\top(\mathbf{1}_T - \boldsymbol{u}) = \frac{1}{T}\mathbf{1}_T \in \mathbb{R}^T$$

$$\boldsymbol{\nu} = \frac{1}{N}\mathbf{1}_N \in \mathbb{R}^N,$$

which represent (respectively) the empirical probability measure of the quantile levels $\boldsymbol{u}$ and the data points $(\boldsymbol{X}, \boldsymbol{y})$ (we choose both measures to be uniform). Now, by using the decomposed objective, and by multiplying the first two constraints by $\boldsymbol{D}^\top$ on either side, we obtain the following equivalent problem:

$$\max_{\boldsymbol{\Pi} \geq 0} \boldsymbol{u}^\top \boldsymbol{\Pi} \boldsymbol{y}$$

$$\text{s.t.} \quad \boldsymbol{\Pi}\mathbf{1}_N = \boldsymbol{\mu} = \frac{1}{T}\mathbf{1}_T \qquad\qquad [\boldsymbol{D}^{-\top}\boldsymbol{a}]$$

$$\boldsymbol{\Pi}\boldsymbol{X} = \boldsymbol{\mu}\boldsymbol{\nu}^\top \boldsymbol{X} = \frac{1}{T}\mathbf{1}_T\bar{\boldsymbol{x}}^\top \quad [\boldsymbol{D}^{-\top}\boldsymbol{B}] \qquad (14)$$

$$\mathbf{1}_T{}^\top \boldsymbol{\Pi} \leq \boldsymbol{\nu} = \frac{1}{N}\mathbf{1}_N{}^\top.$$

Note that (i) we ignore the normalization constants where they do not affect the solution; (ii) the Lagrange multipliers are scaled by $\boldsymbol{D}^{-\top}$ since the constraints are scaled by $\boldsymbol{D}^\top$.

The interpretation of this formulation is that we seek a transport plan $\mathbf{\Pi}$ between the measures of the quantile levels U and the target Y$|$X, subject to constraints on the plan which ensure that its marginals are the empirical measures $\boldsymbol{\mu}$ and $\boldsymbol{\nu}$, and that mean independence $\mathbb{E}\left[\mathbf{X}|\mathrm{U}\right] = \mathbb{E}\left[\mathbf{X}\right]$ holds. Each individual entry $\mathbf{\Pi}_{i,j}$ in this discrete plan is the probability mass attached to $(u_i, \boldsymbol{x}_j, y_j)$ in this optimal joint distribution.

### A.3 EXTENDING THE OPTIMAL TRANSPORT FORMULATION TO VECTOR QUANTILES

We now wish to deal with the case where the target variable can be a vector. Observe that for the scalar case, we can write the OT objective as

$$\boldsymbol{u}^\top \mathbf{\Pi} \boldsymbol{y} = \sum_{i=1}^{T} \sum_{j=1}^{N} \Pi_{i,j} u_i y_j = \mathbf{\Pi} \odot \boldsymbol{S},$$

where $\boldsymbol{S} \in \mathbb{R}^{T \times N}$ is a matrix of pairwise products, i.e. $\boldsymbol{S}_{i,j} = u_i y_j$, and $\odot$ denotes the Hadamard (elementwise) product.

Now let us assume that $\boldsymbol{y}_j \in \mathbb{R}^d$ for any $d \geq 1$, and thus our target data will now be arranged as $\boldsymbol{Y} \in \mathbb{R}^{N \times d}$. The quantile levels must now also $d$-dimensional, since we have a quantile level dimension for each data dimension. We will choose a uniform grid on $[0, 1]^d$ on which to compute our vector quantiles. Along each dimension of this grid we sample $T$ equally-spaced points, giving us in total $T^d$ points in $d$ dimensions.

To keep the formulation of the optimization problem two dimensional, we arrange the coordinates of these points as the matrix $\boldsymbol{U} \in \mathbb{R}^{T^d \times d}$. Thus, we can naturally extend the pairwise product matrix $\boldsymbol{S}$ to the multi-dimensional case using a $d$-dimensional inner-product between each point on the quantile level grid $\boldsymbol{U}$ and each target point in $\boldsymbol{Y}$. This yields the simple form $\boldsymbol{S} = \boldsymbol{U}\boldsymbol{Y}^\top$, where $\boldsymbol{S} \in \mathbb{R}^{T^d \times N}$, which can be plugged in to the above formulation (14) and solved directly. Thus we obtain eq. (5).

### A.4 RELAXING THE EXACT OPTIMAL TRANSPORT DUAL FORMULATION

Recall the exact dual formulation of the OT-based primal VQR problem (5):

$$\min_{\boldsymbol{\psi},\boldsymbol{\varphi},\boldsymbol{\beta}} \boldsymbol{\psi}^\top \boldsymbol{\nu} + \boldsymbol{\varphi}^\top \boldsymbol{\mu} + \mathrm{tr}\left(\boldsymbol{\beta}^\top \bar{\boldsymbol{X}}\right)$$

$$\text{s.t. } \forall i, j :$$

$$\varphi_i + \boldsymbol{\beta}_i^\top \boldsymbol{x}_j + \psi_j \geq \boldsymbol{u}_i^\top \boldsymbol{y}_j$$

The above problem has a unique solution (Carlier et al., 2016). Thus, first-order optimality conditions for each $\varphi_i$ yield

$$\varphi_i = \max_j \left\{ \boldsymbol{u}_i^\top \boldsymbol{y}_j - \boldsymbol{\beta}_i^\top \boldsymbol{x}_j - \psi_j \right\}.$$

Substituting the optimal $\varphi_i$ into the dual formulation results in an unconstrained but exact min-max problem:

$$\min_{\boldsymbol{\psi},\boldsymbol{\beta}} \boldsymbol{\psi}^\top \boldsymbol{\nu} + \mathrm{tr}\left(\boldsymbol{\beta}^\top \bar{\boldsymbol{X}}\right) + \sum_{i=1}^{T^d} \mu_i \max_j \left\{ \boldsymbol{u}_i^\top \boldsymbol{y}_j - \boldsymbol{\beta}_i^\top \boldsymbol{x}_j - \psi_j \right\}. \tag{15}$$

We can relax this problem by using a smooth approximation for the $\max$ operator, given by

$$\max_j(\boldsymbol{x}) \approx \varepsilon \log \left( \sum_j \exp \left( \frac{x_j}{\varepsilon} \right) \right).$$

Plugging the smooth approximation into eq. (15) yields the relaxed dual in eq. (7).

### A.5 EQUIVALENCE BETWEEN REGULARIZED PRIMAL AND RELAXED DUAL

Adding an entropic regularization term to the OT-based primal formulation of the VQR problem (5), and converting into a minimization problem yields,

$$\min_{\mathbf{\Pi}} \ \langle \mathbf{\Pi}, -\boldsymbol{S} \rangle + \varepsilon \langle \mathbf{\Pi}, \log \mathbf{\Pi} \rangle$$

$$\text{s.t. } \mathbf{\Pi}^\top \mathbf{1}_{T^d} = \boldsymbol{\nu} \qquad [-\boldsymbol{\psi}]$$

$$\mathbf{\Pi} \mathbf{1}_N = \boldsymbol{\mu} \qquad [-\boldsymbol{\varphi}] \tag{16}$$

$$\mathbf{\Pi} \boldsymbol{X} = \bar{\boldsymbol{X}} \qquad [-\boldsymbol{\beta}]$$

where $\boldsymbol{\nu} \in \mathbb{R}^N$, $\boldsymbol{\mu} \in \mathbb{R}^{T^d}$, $\bar{\boldsymbol{X}} \in \mathbb{R}^{T^d \times k}$ are defined as in section 3. Note that the entropic regularization term can be interpreted as minimization of the KL-divergence between the transport plan $\boldsymbol{\Pi}$ and the product of marginals, i.e., $\mathcal{D}_{\mathrm{KL}}\left(\boldsymbol{\Pi} \,\|\, \boldsymbol{\mu}\boldsymbol{\nu}^\top\right)$.

In order to show the equivalence between this regularized problem and the relaxed dual (7), the key idea is to use the Fenchel-Rockafeller duality theorem (Rockafellar, 1974). This allows us to write the dual of an optimization problem in terms of the convex conjugate functions of its objective. To apply this approach, we reformulate eq. (16) into an unconstrained problem of the following form:

$$\min_{\boldsymbol{W} \in \mathcal{W}} f^*(\mathcal{A}^*\boldsymbol{W}) + g^*(\boldsymbol{W}) = \max_{\boldsymbol{V} \in \mathcal{V}} -f(-\boldsymbol{V}) - g(\mathcal{A}\boldsymbol{V}),$$

where we define a pair of operators $\mathcal{A} : \mathcal{V} \mapsto \mathcal{W}$ and $\mathcal{A}^* : \mathcal{W}^* \mapsto \mathcal{V}^*$ adjoint to each other; $f : \mathcal{V} \mapsto \mathbb{R}$, $f^* : \mathcal{V}^* \mapsto \mathbb{R}$ and $g : \mathcal{W} \mapsto \mathbb{R}$, $g^* : \mathcal{W}^* \mapsto \mathbb{R}$ are pairs of convex conjugate functions. In our problem $\mathcal{W} = \mathcal{W}^*$ and $\mathcal{V} = \mathcal{V}^*$ are the vector spaces $\mathcal{W} = \mathcal{W}^* = \mathbb{R}^{T^d \times N}$ and $\mathcal{V} = \mathcal{V}^* = \mathbb{R}^{T^d} \times \mathbb{R}^N \times \mathbb{R}^{T^d \times k}$.

We define the operator

$$\mathcal{A}^*\boldsymbol{\Pi} = \left(\boldsymbol{\Pi}^\top \boldsymbol{1}_{T^d}, \boldsymbol{\Pi}\boldsymbol{1}_N, \boldsymbol{\Pi}\boldsymbol{X}\right)$$

and an indicator function,

$$i_{\boldsymbol{a}}(\boldsymbol{z}) = \begin{cases} 0, & \boldsymbol{z} = \boldsymbol{a}, \\ \infty, & \boldsymbol{z} \neq \boldsymbol{a}. \end{cases}$$

We can now represent eq. (16) as an unconstrained problem,

$$\min_{\boldsymbol{\Pi}} \; \langle \boldsymbol{\Pi}, -\boldsymbol{S} \rangle + \varepsilon \langle \boldsymbol{\Pi}, \log \boldsymbol{\Pi} \rangle + i_{(\boldsymbol{\nu}, \boldsymbol{\mu}, \bar{\boldsymbol{X}})}\left(\mathcal{A}^*\boldsymbol{\Pi}\right) \tag{17}$$

To derive $\mathcal{A}$, the adjoint operator of $\mathcal{A}^*$, we can write

$$\begin{aligned}
\langle (\boldsymbol{\psi}, \boldsymbol{\varphi}, \boldsymbol{\beta}), \mathcal{A}^*\boldsymbol{\Pi} \rangle_{\mathcal{V}} &= \langle \boldsymbol{\psi}, \boldsymbol{\Pi}^\top \boldsymbol{1}_{T^d} \rangle + \langle \boldsymbol{\varphi}, \boldsymbol{\Pi}\boldsymbol{1}_N \rangle + \langle \boldsymbol{\beta}, \boldsymbol{\Pi}\boldsymbol{X} \rangle \\
&= \mathrm{tr}\left(\boldsymbol{\psi}^\top \boldsymbol{\Pi}^\top \boldsymbol{1}_{T^d}\right) + \mathrm{tr}\left(\boldsymbol{\varphi}^\top \boldsymbol{\Pi}\boldsymbol{1}_N\right) + \mathrm{tr}\left(\boldsymbol{\beta}^\top \boldsymbol{\Pi}\boldsymbol{X}\right) \\
&= \langle \boldsymbol{1}_{T^d}\boldsymbol{\psi}^\top, \boldsymbol{\Pi} \rangle + \langle \boldsymbol{\varphi}\boldsymbol{1}_N^\top, \boldsymbol{\Pi} \rangle + \langle \boldsymbol{\beta}\boldsymbol{X}^\top, \boldsymbol{\Pi} \rangle \\
&= \langle \mathcal{A}(\boldsymbol{\psi}, \boldsymbol{\varphi}, \boldsymbol{\beta}), \boldsymbol{\Pi} \rangle_{\mathcal{W}}.
\end{aligned}$$

Thus, $\mathcal{A}(\boldsymbol{\psi}, \boldsymbol{\varphi}, \boldsymbol{\beta}) = \boldsymbol{1}_{T^d}\boldsymbol{\psi}^\top + \boldsymbol{\varphi}\boldsymbol{1}_N^\top + \boldsymbol{\beta}\boldsymbol{X}^\top$.

We then define the functions

$$\begin{aligned}
f^*(\mathcal{A}^*\boldsymbol{\Pi}) &= i_{(\boldsymbol{\nu}, \boldsymbol{\mu}, \bar{\boldsymbol{X}})}\left(\mathcal{A}^*\boldsymbol{\Pi}\right) \\
g^*(\boldsymbol{\Pi}) &= \langle \boldsymbol{\Pi}, -\boldsymbol{S} \rangle + \varepsilon \langle \boldsymbol{\Pi}, \log \boldsymbol{\Pi} \rangle,
\end{aligned}$$

and their corresponding convex conjugates are therefore given by,

$$\begin{aligned}
f(\boldsymbol{\psi}, \boldsymbol{\varphi}, \boldsymbol{\beta}) &= \langle (\boldsymbol{\psi}, \boldsymbol{\varphi}, \boldsymbol{\beta}), (\boldsymbol{\nu}, \boldsymbol{\mu}, \bar{\boldsymbol{X}}) \rangle_{\mathcal{V}} = \langle \boldsymbol{\psi}, \boldsymbol{\nu} \rangle + \langle \boldsymbol{\varphi}, \boldsymbol{\mu} \rangle + \langle \boldsymbol{\beta}, \bar{\boldsymbol{X}} \rangle \\
g(\boldsymbol{W}) &= \varepsilon \sum_{ij} \exp\left(\frac{W_{ij} + S_{ij} - 1}{\varepsilon}\right).
\end{aligned}$$

Using $f^*$ and $g^*$, we can write (17) as $\min_{\boldsymbol{W}} \{f^*(\mathcal{A}^*\boldsymbol{W}) + g^*(\boldsymbol{W})\}$, then by the Fenchel-Rockafeller duality theorem, we get the equivalent dual form $\max_{\boldsymbol{V}} \{-f(-\boldsymbol{V}) - g(\mathcal{A}\boldsymbol{V})\}$. Substituting $\boldsymbol{V} = (-\boldsymbol{\psi}, -\boldsymbol{\varphi}, -\boldsymbol{\beta})$ (the dual variables of eq. (16)), converting to a minimization problem, and omitting constant factors in the objective, we get

$$\min_{\boldsymbol{\psi}, \boldsymbol{\varphi}, \boldsymbol{\beta}} \boldsymbol{\psi}^\top \boldsymbol{\nu} + \boldsymbol{\varphi}^\top \boldsymbol{\mu} + \mathrm{tr}\left(\boldsymbol{\beta}^\top \bar{\boldsymbol{X}}\right) + \varepsilon \sum_{i=1}^{T^d} \sum_{j=1}^{N} \exp\left(\frac{1}{\varepsilon}\left(S_{ij} - \psi_j - \varphi_i - \boldsymbol{\beta}_i^\top \boldsymbol{x}_j\right)\right). \tag{18}$$

Now we write $\boldsymbol{\varphi}$ in terms of $\boldsymbol{\psi}, \boldsymbol{\beta}$ by using a first-order optimality condition of the above problem. Taking the derivative w.r.t. $\varphi_i$ and setting to zero, we have:

$$0 = \mu_i - \exp\left(-\frac{\varphi_i}{\varepsilon}\right) \sum_{j=1}^{N} \exp\left(\frac{1}{\varepsilon}\left(S_{ij} - \psi_j - \boldsymbol{\beta}_i^\top \boldsymbol{x}_j\right)\right)$$

$$\exp\left(-\frac{\varphi_i}{\varepsilon}\right) = \frac{\mu_i}{\sum_j \exp\left(\frac{1}{\varepsilon}\left(S_{ij} - \psi_j - \boldsymbol{\beta}_i^\top \boldsymbol{x}_j\right)\right)} \tag{19}$$

$$\varphi_i = \varepsilon \log\left(\frac{1}{\mu_i} \sum_j \exp\left(\frac{1}{\varepsilon}\left(S_{ij} - \psi_j - \boldsymbol{\beta}_i^\top \boldsymbol{x}_j\right)\right)\right) \tag{20}$$

Finally, substituting eqs. (19) and (20) into eq. (18) and omitting constant terms yields,

$$\min_{\boldsymbol{\psi},\boldsymbol{\beta}} \ \boldsymbol{\psi}^\top \boldsymbol{\nu} + \operatorname{tr}\left(\boldsymbol{\beta}^\top \bar{\boldsymbol{X}}\right) + \varepsilon \sum_{i=1}^{T^d} \mu_i \log\left(\sum_{j=1}^{N} \exp\left(\frac{1}{\varepsilon}\left(S_{ij} - \boldsymbol{\beta}_i^\top \boldsymbol{x}_j - \psi_j\right)\right)\right), \tag{21}$$

where $S_{ij} = \boldsymbol{u}_i^\top \boldsymbol{y}_j$. Thus, we obtain that eq. (21) is equal to eq. (7).

In summary, we have shown that the relaxed dual formulation of the VQR problem that we solve (7), is equivalent to an entropic-regularized version of the OT-based primal formulation eq. (5).

## A.6 EXTRACTING THE VECTOR QUANTILE REGRESSION COEFFICIENTS

The dual variables obtained from the OT formulation (eq. (4)) are $\boldsymbol{\varphi} \in \mathbb{R}^{T^d}$ and $\boldsymbol{\beta} \in \mathbb{R}^{T^d \times k}$. In the case of scalar quantiles ($d = 1$) we can obtain the conditional quantile function from the dual variables by applying the $T \times T$ finite differences matrix $\boldsymbol{D}^\top$ defined in appendix A.1:

$$\widehat{Q}_{Y|\mathbf{X}}(u; \boldsymbol{x}) = \left[\boldsymbol{D}^\top\left(\boldsymbol{\beta}\boldsymbol{x} + \boldsymbol{\varphi}\right)\right]_u.$$

This is effectively taking the $u$-th component the first-order discrete derivative where $u$ is one of the $T$ discrete quantile levels.

In the vector case, we have $T^d$ discrete $d$-dimensional quantile levels. Equivalently, the relation between the dual variables and the quantile function is then

$$\widehat{Q}_{\mathbf{Y}|\mathbf{X}}(\boldsymbol{u}'; \boldsymbol{x}) = \left[\nabla_{\boldsymbol{u}}\left\{\boldsymbol{\beta}\boldsymbol{x} + \boldsymbol{\varphi}\right\}\right]_{\boldsymbol{u}'}.$$

Here $\boldsymbol{\beta}\boldsymbol{x} + \boldsymbol{\varphi}$ is in $\mathbb{R}^{T^d}$ and its gradient with respect to $\boldsymbol{u}$, $\nabla_{\boldsymbol{u}}\left\{\boldsymbol{\beta}\boldsymbol{x} + \boldsymbol{\varphi}\right\}$, is in $\mathbb{R}^{T^d \times d}$. We then evaluate its gradient at o ne of the discrete levels $\boldsymbol{u}'$ to obtain the $d$-dimensional quantile. As explained in section 3, the expression $\boldsymbol{\beta}\boldsymbol{x} + \boldsymbol{\varphi}$ is convex in $u$, and thus the co-monotonicity of the estimated CVQF is obtained by virtue of it being the gradient of a convex function.

## B    CONVERGENCE OF VQR

Below we mention a few comments regarding the optimization and approximation error of VQR.

**Linear VQR.** The relaxed dual formulation of VQR, presented in eq. (6), is an unconstrained smooth convex minimization problem. Thus, in this case, gradient descent is guaranteed to converge to the optimal solution up to any desired precision. Moreover, given fixed quantile levels, our objective can be written as the minimization of an expectation under the data distribution $P_{(\mathbf{X},\mathbf{Y})}$. Under these criteria, performing SGD by sampling i.i.d. from the data distribution, is known to converge to the global minimum (Hazan (2019), Chapter 3). We refer to Section 5.4 in (Peyré & Cuturi, 2019) (and references therein) for further analysis and details regarding the convergence rates of SGD and other stochastic optimization methods applied to this problem. Specifically, stochastic averaged gradient (SAG) was shown to have improved convergence rates compared to SGD Genevay et al. (2016). However, in practice we find that SGD converges well for this problem (see Figure 2, Appendix fig. A10a), and we use it here for simplicity.

**Nonlinear VQR.** In the case of non-linear VQR, as formulated in eq. (8), the optimization is over a non-convex objective, and thus no convergence to a global minimum is guaranteed. Convergence analyses for non convex objectives only provide guarantees for weak forms of convergence. For e.g., the analysis methodology introduced by Ghadimi & Lan (2013) can be used to show that under the assumption of uniformly bounded gradient estimates, the norm of the gradient of the loss function decreases on average as $\mathcal{O}(t^{-1/2})$, when $t \to \infty$, where $t$ is the iteration. This can be viewed as a weak form of convergence that does not guarantee convergence to a fixed point, even to a local minimum.

**Approximation error as $\varepsilon \to 0$.** The nonlinear VQR formulation (8) produces an estimate $\widehat{Q}^{\varepsilon}_{\mathbf{Y}|\mathbf{X}}$. By decreasing $\varepsilon$, one can make the problem more exact, as we have shown in practice (fig. A10b). Following the approach of Proposition A.1 in Genevay et al. (2016), it can be shown that this estimate approaches the true CVQF as $\varepsilon \to 0$.

## C  CONDITIONAL VECTOR QUANTILE FUNCTIONS

### C.1  INTUITIONS ABOUT VECTOR QUANTILES

To interpret the meaning of CVQFs, let us consider the 2-dimensional case, where we assume $\mathbf{Y} = (Y_1, Y_2)^\top$. Given a specific covariates vector $\boldsymbol{x} = (x_1, \ldots, x_k)^\top$, and level $\boldsymbol{u} = (u_1, u_2)^\top$ we may write the components of the conditional vector quantile function as,

$$Q_{\mathbf{Y}|\mathbf{X}}(\boldsymbol{u}; \boldsymbol{x}) = \begin{bmatrix} Q_1(\boldsymbol{u}; \boldsymbol{x}) \\ Q_2(\boldsymbol{u}; \boldsymbol{x}) \end{bmatrix} = \begin{bmatrix} Q_{Y_1|Y_2,\mathbf{X}}\left(u_1; Q_{Y_2|\mathbf{X}}\left(u_2; \boldsymbol{x}\right), \boldsymbol{x}\right) \\ Q_{Y_2|Y_1,\mathbf{X}}\left(u_2; Q_{Y_1|\mathbf{X}}\left(u_1; \boldsymbol{x}\right), \boldsymbol{x}\right) \end{bmatrix},$$

where $Q_{Y_i|Y_j,\mathbf{X}}(u; y, \boldsymbol{x})$ denotes the scalar quantile function of the random variable $Y_i$ at level $u$, given $Y_j = y, \mathbf{X} = \boldsymbol{x}$. Thus, for example, the first component $Q_1(\boldsymbol{u}; \boldsymbol{x})$ is a 2D surface where moving along $u_1$ for a fixed $u_2$ yields a scalar, non-decreasing function representing the quantiles of $Y_1$ when $Y_2$ is at a value corresponding to level $u_2$ (see Figure A6b). In addition, the vector quantile function is co-monotonic with $\boldsymbol{u}$ in the sense defined by eq. (1). For higher dimensions, it becomes more involved as the conditioning in each component is on the vector quantile of all the remaining components of the target $\mathbf{Y}$ (Figure A6c).

### C.2  CONDITIONAL QUANTILE CONTOURS

A useful property of CVQFs is that they allow a natural extension of $\alpha$-confidence intervals to high dimensional distributions, which we denote as $\alpha$-contours.

**Vector quantile contours.**  Formally, we define the conditional contour of $\mathbf{Y}|\mathbf{X} = \boldsymbol{x}$ at confidence level $\alpha$ as $\mathcal{Q}_{\mathbf{Y}|\mathbf{X}}^\alpha(\boldsymbol{x})$, given by

$$\mathcal{Q}_{\mathbf{Y}|\mathbf{X}}^\alpha(\boldsymbol{x}) = \left\{ Q_{\mathbf{Y}|\mathbf{X}}(\boldsymbol{u}; \boldsymbol{x}) \mid \boldsymbol{u} \in \mathcal{U}_\alpha \right\}, \tag{22}$$

$$\mathcal{U}_\alpha = \bigcup_{i=1}^{d} \mathcal{U}_\alpha^i,$$

$$\mathcal{U}_\alpha^i = \left\{ (u_1, \ldots, u_d)^\top \mid u_i \in [\alpha, 1 - \alpha], \ u_{-i} \in \{\alpha, 1 - \alpha\} \right\}$$

where $u_{-i}$ denotes any component of $\boldsymbol{u} = (u_1, \ldots, u_d)^\top$ except for $u_i$. Note that the contour can be calculated not only using the true CVQF, $Q_{\mathbf{Y}|\mathbf{X}}(\boldsymbol{u}; \boldsymbol{x})$, as above, but also using an estimated CVQF, $\widehat{Q}_{\mathbf{Y}|\mathbf{X}}(\boldsymbol{u}; \boldsymbol{x})$.

For simplicity consider the 2d case. By fixing e.g. $u_1$ to be one of $\{\alpha, 1 - \alpha\}$ and then sweeping over $u_2$ (and vice versa), we obtain a set of vector quantile levels $\mathcal{U}_\alpha$ corresponding to the confidence level $\alpha$ (see fig. 1a, left and right). Mapping the set $\mathcal{U}_\alpha$ back to the domain of $\mathbf{Y}$ by using the values of the CVQF $Q_{\mathbf{Y}|\mathbf{X}}(\boldsymbol{u}; \boldsymbol{x})$ along it, we obtain a contour of arbitrary shape, which contains $100 \cdot (1 - 2\alpha)^d$ percent of the $d$-dimensional distribution (see fig. 1a, middle and fig. A7, right).

**Separable quantile contours.**  In contrast to vector quantile contours obtained by VQR, which can accurately model distributions with arbitrary shapes, using separable quantiles produces trivial box-shaped contours. Consider a CVQF estimated with separable quantiles as presented in eq. (10). Due to the dependence of each component of $\widehat{Q}_{\mathbf{Y}|\mathbf{X}}^{\text{Sep}}(\boldsymbol{u}; \boldsymbol{x})$ only on the corresponding component of $\boldsymbol{u}$, the shape of the resulting quantile contour will always be box-shaped (see fig. A7, middle). Such trivial contours result in inferior performance for applications of uncertainty estimation, where a confidence region with a the smallest possible area for a given confidence level is desired.

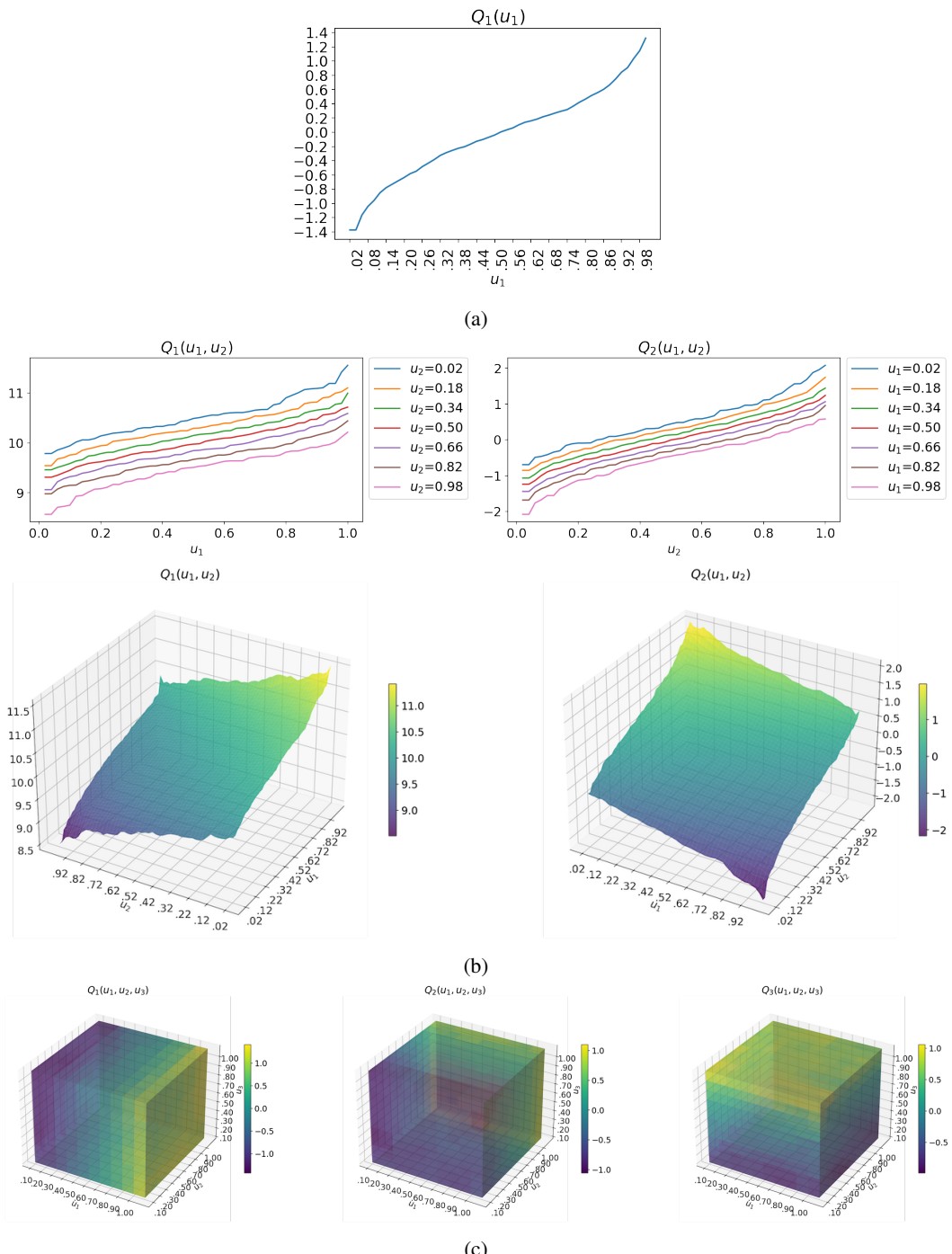

Figure A6: **Visualization (a) 1d, (b) 2d, and (c) 3d vector quantile functions estimated on the MVN dataset.** In each plot, $Q_i(\boldsymbol{u})$ is the $i$th component of the vector quantile $\boldsymbol{Q_Y}(\boldsymbol{u})$, plotted over all quantile levels $\boldsymbol{u}$. The number of quantile levels calculated was 50, 25 and 10 for 1d, 2d and 3d quantiles respectively. For 2d quantiles (b), the top plot shows multiple monotonic quantile curves of one variable while keeping the other at a fixed level.

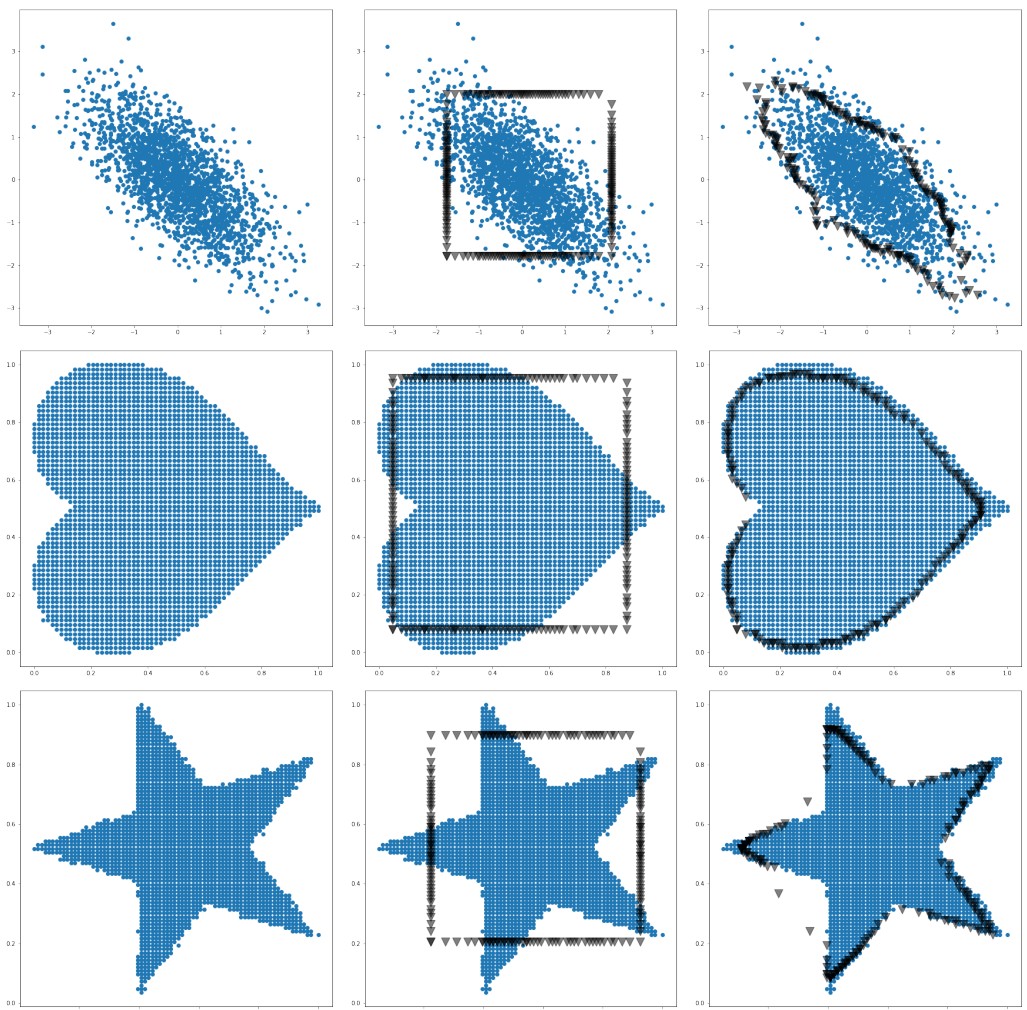

Figure A7: **Visualization of $\alpha$-quantile contours constructed using separable quantiles and vector quantiles**. All the contours presented contain $\sim 90\%$ of the points (i.e., $\alpha = 0.025$). Left (top to bottom): samples drawn from bivariate normal, heart-shaped, and star-shaped densities. Middle & Right: black triangles overlaid on the density constitute the $\alpha-$quantile contour estimated using separable quantiles and vector quantiles, respectively. The bivariate normal density has zero mean, unit variance and correlation of $\rho = -0.7$. Vector quantile contours accurately capture the shape of distribution whereas the separable quantile contours are always box-shaped due to the assumption of independence.

# D    DATASETS

## D.1    SYNTHETIC DATASETS

Below we describe the four synthetic datasets which were used in the experiments.

**MVN.** Data was generated from a linear model $\boldsymbol{y} = \boldsymbol{A}\boldsymbol{x} + \boldsymbol{\eta}$, where $\boldsymbol{x} \sim \mathbb{U}[0,1]^k$, $\boldsymbol{A} \in \mathbb{R}^{d \times k}$ is a random projection matrix and $\boldsymbol{\eta}$ is multivariate Gaussian noise with a random covariance matrix.

**Conditional-banana.** This dataset was introduced by Feldman et al. (2021), and inspired by a similar dataset in Carlier et al. (2017). The target variable $\mathbf{Y} \in \mathbb{R}^2$ has a banana-shaped conditional distribution, and its shape changes non-trivially when conditioned on a continuous-valued covariate $\mathrm{X} \in \mathbb{R}$ (see the left-most column of the panel presented in Figure A13). The data generating process is defined as follows

$$\mathrm{X} \sim \mathbb{U}[0.8, 3.2], \ \ \mathrm{z} \sim \mathbb{U}[-\pi, \pi], \ \ \phi \sim \mathbb{U}[0, 2\pi], \ \ \mathrm{r} \sim \mathbb{U}[-0.1, 0.1]$$

$$\hat{\beta} \sim \mathbb{U}[0,1]^k, \ \ \beta = \frac{\hat{\beta}}{||\hat{\beta}||_1}$$

$$\mathrm{Y}_0 = \frac{1}{2}\left(-\cos\left(\mathbf{Z}\right) + 1\right) + r\sin\left(\phi\right) + \sin\left(\mathrm{X}\right), \ \ \mathrm{Y}_1 = \frac{\mathrm{Z}}{\beta\mathrm{X}} + \mathrm{r}\cos\left(\phi\right),$$

and then $\mathbf{Y} = [\mathrm{Y}_0, \mathrm{Y}_1]^\top$.

**Synthetic glasses.** This dataset was introduced by Brando et al. (2022). The target variable $\mathrm{Y} \in \mathbb{R}$ has a bimodal conditional distribution. The mode locations shift periodically when conditioned on a continuous-valued covariate $\mathrm{X} \sim \mathbb{U}[0,1]$ (fig. A12a). The data generating process is defined as

$$
\begin{aligned}
\mathrm{z}_1 &= 3\pi\mathrm{X} & \mathrm{z}_2 &= \pi(1 + 3\mathrm{X}) \\
\epsilon &\sim \mathrm{Beta}\left(\alpha = 0.5, \beta = 1\right) & \gamma &\sim \mathrm{Categorical}(0, 1) \\
\mathrm{Y}_1 &= 5\sin(\mathrm{z}_1) + 2.5 + \epsilon & \mathrm{Y}_2 &= 5\sin(\mathrm{z}_2) + 2.5 - \epsilon \\
\mathrm{Y} &= \gamma\mathrm{Y}_1 + (1 - \gamma)\mathrm{Y}_2. &&
\end{aligned}
$$

**Rotating star.** In this dataset, the target variable of $\mathbf{Y} \in \mathbb{R}^2$ has a star-shaped conditional distribution, and its shape is rotated by $\mathrm{X} \in \mathbb{R}$ degrees when conditioned on a discrete-valued X, taking values in $[0, 10, 20, \ldots, 60]$. Data is generated based on a $600 \times 600$ binary image of a star. See the first column in Figure A14 to visualize conditional distributions as a function of X. Since conditional distributions differ only by a rotation, $\mathbb{E}\left[\mathbf{Y}|\mathrm{X}\right]$ remains the same for all X. However, the shape of the distribution changes substantially with X, especially in tails. Thus, this dataset is a challenging candidate for estimating vector quantiles which must also represent these tails in order to properly recover the shape of the conditional distribution.

## D.2    REAL DATASETS

We perform real data experiments on the `blog_data`, `bio`, `house`, `meps_20` datasets obtained from (Feldman et al., 2021). The original real datasets contained one-dimensional targets. Feldman et al. (2021) constructed an additional target variable by selecting a feature that has high correlation with first target variable and small correlation to the other input features, so that it is hard to predict. Summary of these datasets is presented in table A1.

| Dataset Name | $N$ | $k$ | $d$ | Additional Response |
|---|---|---|---|---|
| `blog_data` | 52397 | 279 | 2 | Time between the blog post publication and base-time |
| `bio` | 45730 | 8 | 2 | F7 - Euclidean distance |
| `house` | 21613 | 17 | 2 | Latitude of a house |
| `meps_20` | 17541 | 138 | 2 | Overall rating of feelings |

Table A1: Information about the real data sets.

# E  EVALUATION METRICS

## E.1  METRICS FOR SYNTHETIC DATA

The quality of an estimated CVQF can be measured by how well it upholds strong representation and co-monotonicity (eqs. (1) and (2)). However, measuring strong representation requires knowledge of the ground-truth joint distribution $P_{(\mathbf{X},\mathbf{Y})}$ or its conditional quantile function. With synthetic data, adherence to strong representation can be measured via several proxy metrics. The second key property, violations of co-monotonicity, can be measured directly. Below we describe the metrics we use to evaluate these properties on estimated conditional quantile functions.

**Entropy of inverse CVQF.**  If strong representation holds, the inverted CVQF, $\widehat{Q}^{-1}_{\mathbf{Y}|\mathbf{X}}(\mathbf{Y}|\mathbf{X})$, must result in a uniform distribution, when evaluated on samples drawn from the true conditional distribution. As a measure of uniformity, we calculate a normalized entropy of the inverted CVQF. The inversion procedure is done as follows.

1. Sample $L$ evaluation covariate vectors, $\{\boldsymbol{x}_l\}_{l=1}^{L}$ values at random from the ground truth distribution $P_{\mathbf{X}}$.
2. For each $\boldsymbol{x}_l$,
   (a) Sample $M$ points $\{\boldsymbol{y}_{m,l}\}_{m=1}^{M}$ from $\mathbf{Y}|\mathbf{X} = \boldsymbol{x}_l$.
   (b) For each $\boldsymbol{y}_{m,l}$, find the level of the closest vector quantile w.r.t. the Euclidean distance, i.e.
   $$\boldsymbol{u}_{m,l} = \arg\min_{\boldsymbol{u}'} \left\| \boldsymbol{y}_m - \widehat{Q}_{\mathbf{Y}|\mathbf{X}}(\boldsymbol{u}'; \boldsymbol{x}_l) \right\|_2.$$
   (c) For $i \in \{1, \ldots, T^d\}$ denote $c_i$ as the number of times that the quantile level $i$ is found in $\{\boldsymbol{u}_{m,l}\}_{m=1}^{M}$, and calculate $p_i = c_i/M$.
   (d) Calculate the entropy $h_l = -\sum_i p_i \log p_i$ and the normalized entropy $\widetilde{h}_l = (\exp(h_l) - 1)/(T^d - 1)$.
3. Report the CVQF inverse entropy as $\frac{1}{L}\sum_l \widetilde{h}_l$.

Note that,

1. The entropy metric is normalized such that its values are in $[0, 1]$ where 1 corresponds to a uniform distribution, and 0 corresponds to a delta distribution.
2. Some non-uniformity arises due to the quantization of the quantile level grid into $T$ discrete levels per dimension. We report a reference entropy, calculated on a uniform sample of $M$ quantile-level grid points.
3. We used $L = 20$ and $M = 4000$.

**Distribution distance (KDE-L1).**  Since the CVQF fully represents the conditional distribution $P_{(\mathbf{Y}|\mathbf{X})}$ it can serve as a generative model for it. We used inverse-transform sampling to generate data from the estimated conditional distribution though the fitted VQR model, $\widehat{Q}_{\mathbf{Y}|\mathbf{X}}(\boldsymbol{u}; \boldsymbol{x})$, as follows.

1. Sample $L$ evaluation covariate vectors, $\{\boldsymbol{x}_l\}_{l=1}^{L}$ values at random from the ground truth distribution $P_{\mathbf{X}}$.
2. For each $\boldsymbol{x}_l$,
   (a) Sample $M$ quantile levels $\{\boldsymbol{u}_{m,l}\}_{m=1}^{M}$ uniformly.
   (b) Generate $M$ samples from the estimated distribution of $\mathbf{Y}|\mathbf{X}$ using the estimated CVQF: $\left\{\hat{\boldsymbol{y}}_{m,l} = \widehat{Q}_{\mathbf{Y}|\mathbf{X}}(\boldsymbol{u}_{m,l}; \boldsymbol{x}_l)\right\}_{m=1}^{M}$.
   (c) Sample an additional $M$ points $\left\{\boldsymbol{y}^*_{m,l}\right\}_{m=1}^{M}$ from the ground truth conditional distribution $P_{\mathbf{Y}|\mathbf{X}=\boldsymbol{x}_l}$.
   (d) Calculate a Kernel Density Estimate (KDE) $\widehat{f}_{\mathbf{Y}|\mathbf{X}=\boldsymbol{x}_l}$ from the VQR samples $\{\hat{\boldsymbol{y}}_{m,l}\}$.

(e) Calculate the KDE $f^*_{\mathbf{Y}|\mathbf{X}=\boldsymbol{x}_l}$ from the ground truth samples $\left\{\boldsymbol{y}^*_{m,l}\right\}$.

3. Calculate the KDE-L1 metric as

$$\frac{1}{L} \sum_{l=1}^{L} \left\| f^*_{\mathbf{Y}|\mathbf{X}=\boldsymbol{x}_l} - \widehat{f}_{\mathbf{Y}|\mathbf{X}=\boldsymbol{x}_l} \right\|_1 .$$

The KDEs are calculated with 100 bins per dimension. An isotropic Gauassian kernel was used, with $\sigma = 0.1$ for the conditional-banana dataset and $\sigma = 0.035$ for the star dataset. We used `pykeops` (Charlier et al., 2021) for a fast implementation of high-dimensional KDEs. We used $L = 20$ and $M = 4000$.

**Quantile function distance (QFD).** This metric measures the distance between a true CVQF, $Q^*_{\mathbf{Y}|\mathbf{X}}$, and an estimate for it obtained by VQR, $\widehat{Q}_{\mathbf{Y}|\mathbf{X}}$.

1. Sample $L$ evaluation covariate vectors, $\{\boldsymbol{x}_l\}_{l=1}^{L}$ values at random from the ground truth distribution $P_{\mathbf{X}}$.

2. For each $\boldsymbol{x}_l$,

   (a) Sample $M$ points $\{\boldsymbol{y}_{m,l}\}_{m=1}^{M}$ from the ground truth conditional distribution $P_{\mathbf{Y}|\mathbf{X}=\boldsymbol{x}_l}$.

   (b) Estimate an unconditional vector quantile function on $\{\boldsymbol{y}_{m,l}\}_{m=1}^{M}$, i.e. perform vector quantile estimation, not regression. Denote the estimated unconditional vector quantile function as $Q^*_{\mathbf{Y}|\mathbf{X}}$. This serves as a proxy for the ground-truth conditional quantile function.

   (c) Denote the estimated conditional quantile function evaluated at $\boldsymbol{x}_l$: $\widehat{Q}_{\mathbf{Y}|\mathbf{X}=\boldsymbol{x}_l}$.

   (d) Compute the normalized difference between them elementwise over each of the $T^d$ discrete quantile levels, i.e.,

   $$d_l = \left\| Q^*_{\mathbf{Y}|\mathbf{X}=\boldsymbol{x}_l} - \widehat{Q}_{\mathbf{Y}|\mathbf{X}=\boldsymbol{x}_l} \right\|_2 \Big/ \left\| Q^*_{\mathbf{Y}|\mathbf{X}=\boldsymbol{x}_l} \right\|_2 .$$

3. Calculate the QFD metric as $\frac{1}{L} \sum_{l=1}^{L} d_l$.

We used $L = 20$ and $M = 4000$.

**Percentage of co-monotonicity violations (MV).** This value can be measured directly. Given an estimated vector quantile function $\widehat{Q}(\boldsymbol{u})$, with $T$ levels per dimension, there are in total $T^{2d}$ quantile level *pairs*. Thus, we measure

$$\frac{1}{T^{2d}} \sum_{i,j}^{T^d} \mathbb{I}\left\{ (\boldsymbol{u}_i - \boldsymbol{u}_j)^\top (\widehat{Q}(\boldsymbol{u}_i) - \widehat{Q}(\boldsymbol{u}_j)) < 0 \right\} .$$

### E.2 METRICS FOR REAL DATA

With real data, only finite samples from the ground truth distribution are available. In particular, since our real datasets have a continuous $\mathbf{X}$, this means that we never have more than one sample from each conditional distribution $\mathbf{Y}|\mathbf{X} = \boldsymbol{x}$. Strong representation can therefore not be quantified directly as with the above metrics. Instead, we opt for metrics which can be evaluated on finite samples and are specifically suitable for our chosen application of distribution-free uncertainty estimation (section 6.3).

**Size of a conditional $\alpha$-confidence set (AC).** This metric approximates $|\mathcal{C}_\alpha(\boldsymbol{x})|$ i.e. the size of an $\alpha$-confidence set constructed for a specific covariate $\boldsymbol{x}$. Lower values of this metric are better, as they indicate that the confidence set is a better fit for the shape of the data distribution because, intuitively, the implication is that the same proportion of the data distribution can be represented by a smaller region. The metric is computed as follows.

1. Estimate a CVQF $\widehat{Q}_{\mathbf{Y}|\mathbf{X}}(\boldsymbol{u}; \boldsymbol{x})$, e.g. via VQR (7), NL-VQR (8), or separable quantiles (10).

2. Choose a test covariate, denoted by $\boldsymbol{x}^T$.

3. Construct the corresponding conditional $\alpha$-contour, $\mathcal{Q}^{\alpha}_{\mathbf{Y}|\mathbf{X}}(\boldsymbol{x}^T)$, as defined by eq. (22), using the estimate $\widehat{Q}_{\mathbf{Y}|\mathbf{X}}(\boldsymbol{u}; \boldsymbol{x}^T)$.

4. Construct the corresponding conditional $\alpha$-confidence set $\mathcal{C}_{\alpha}(\boldsymbol{x}^T)$ from the contour $\mathcal{Q}^{\alpha}_{\mathbf{Y}|\mathbf{X}}(\boldsymbol{x}^T)$ by calculating a convex hull of the points within the contour.

5. Calculate the value of the metric as the volume of the resulting convex hull (area for $d = 2$).

We note that using a convex hull is only a linear approximation of the true $\mathcal{C}_{\alpha}(\boldsymbol{x})$ defined by the points in the contour. For reasonable values of $T$, we find it to be a good approximation, and it is an upper-bound on the area/volume of the true $\mathcal{C}_{\alpha}(\boldsymbol{x})$. In practice we use `scipy` with `qhull` (Virtanen et al., 2020; Barber et al., 1996) to construct $d$-dimensional convex hulls and measure their volume.

**Marginal Coverage (MC).** This metric measures the proportion of unseen data points that are contained within the the conditional $\alpha$-confidence sets, $\mathcal{C}_{\alpha}(\boldsymbol{x})$, that was obtained from a given estimated CVQF. It is computed as follows.

1. Estimate a CVQF $\widehat{Q}_{\mathbf{Y}|\mathbf{X}}(\boldsymbol{u}; \boldsymbol{x})$ using one of the aforementioned methods on a training set sampled from $P_{\mathbf{Y}|\mathbf{X}}$.

2. Denote a disjoint held-out test set

$$\left\{\boldsymbol{x}^T_j, \boldsymbol{y}^T_j\right\}^{N_T}_{j=1} \sim P_{\mathbf{Y}|\mathbf{X}}.$$

3. Measure the marginal coverage as

$$\frac{1}{N_T} \sum_j^{N_T} \mathbb{I}\left\{\boldsymbol{y}^T_j \in \mathcal{C}_{\alpha}(\boldsymbol{x}^T_j)\right\},$$

where $\mathcal{C}_{\alpha}(\boldsymbol{x}^T_j)$ is constructed as a convex hull of the conditional $\alpha$-contour $\mathcal{Q}^{\alpha}_{\mathbf{Y}|\mathbf{X}}(\boldsymbol{x}^T_j)$ constructed using the estimated $\widehat{Q}_{\mathbf{Y}|\mathbf{X}}(\boldsymbol{u}; \boldsymbol{x})$ (eq. (22)).

# F    ADDITIONAL EXPERIMENTS

## F.1    SCALE

**Scale with respect to** $d$ **and** $k$. Figure A8 demonstrates the runtime of our solver with respect to the number of target dimensions $d$ and covariate dimensions $k$.

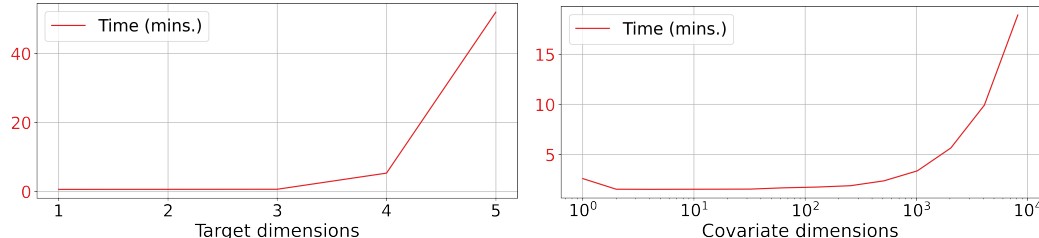

Figure A8:  **Effect of** $d$ **and** $k$ **on runtime.**  VQR solver runtime is shown as a function of number of target dimensions $d$ (left) and covariate dimensions $k$ (right). Calculated on the MVN dataset with $N = 10k$; for $d$ experiment $T = 10$; for $k$ experiment $T = 50$ and $d = 2$. Runtime scales exponentially with $d$ as expected, due to having $T^d$ quantile levels. For $k$, we can see that runtime remains relatively constant even for hundreds of dimensions, then increses due to memory constraints.

**Comparison to general-purpose solver.** To showcase the scalability of our solver, we compare its runtime against an off-the-shelf linear program solver, ECOS, available within the `CVXPY` package (Diamond & Boyd, 2016). These results are shown in Figure A9. As expected, our custom VQR solver approaches the performance of the general purpose solver which achieves slightly more accurate solutions due to solving an exact problem. However, the results demonstrate that with a linear program solver, the runtime quickly becomes prohibitive even for small problems. Crucially, it is important to note that linear program solvers only allow solving the linear VQR problem, while our solver supports nonlinear VQR.

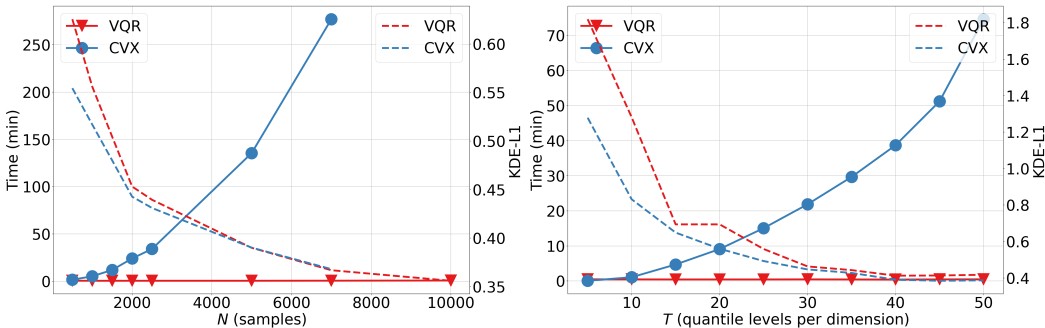

Figure A9:    **Our proposed solver is orders of magnitude faster than general-purpose linear program solver, and approaches the exact solution as the problem size increases.**  VQR and CVX (using ECOS LP solver) solver runtimes (solid lines) are shown as a function of number of samples $N$ (left) and quantile levels per dimension $T$ (right). Dashed lines indicate KDE-L1 as measure of solution quality. CVX obtains the ideal solution due to solving an exact problem. Calculated on the MVN dataset $d = 2$, $k = 10$. Left: $T = 30$; Right: $N = 2000$.

## F.2    OPTIMIZATION

To further evaluate our proposed VQR solver, we experimented with various values of $\varepsilon$ and batch sizes of samples and quantile levels, denoted as $B_N$ and $B_T$ respectively. Figure A10a shows the effect of the mini-batch size in both $N$ and $T$ on the accuracy of the solution. We measured the QFD metric, averaged over 100 evaluation $\boldsymbol{x}$ values. These results are presented in Figure A10b. As expected we observe improved accuracy when increasing batch sizes (both $B_N$ and $B_T$) and when decreasing $\varepsilon$.

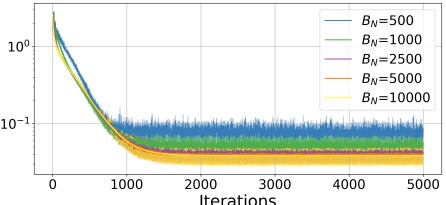 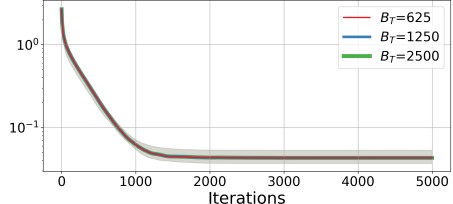

(a) **Effect of batch sizes $B_N$ and $B_T$ on optimization.** Computed on MVN dataset with $d = 2$ and $k = 1$. The vertical axis is the QFD metric defined in section 6. Left: Sweeping over $B_N$; $T = 25$. Right: Sweep over $B_T$; $T = 50$ produces $50^2$ total levels.

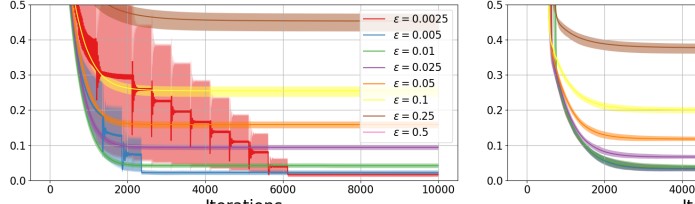

(b) **Decreasing $\varepsilon$ in the relaxed dual improves strong representation.** Y-axis: QFD metric. Computed on MVN dataset with $N = 20k$, $k = 1$. Right: $T = 50$, $d = 2$; Left: $T = 100$, $d = 1$.

Figure A10: Optimization experiments showing the effect of $\varepsilon$ and both batch sizes on convergence.

### F.3 VMR

Figure 4 shows the effectiveness of our VMR procedure. We can see that when applying VMR, there are no monotonicity violations, even for low values of $\varepsilon$, and, moreover, QFD metric improves. Figures A12b and A12c demonstrate how VMR can remove monotonicity violations, even when the distribution cannot be estimated correctly (in this case due to linear VQR). Finally, fig. A11 visualizes the co-monotonicity violations in 2d.

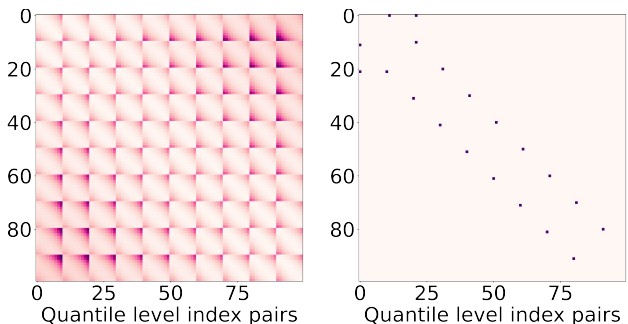

Figure A11: **Visualization of monotonicity violations in 2d.** Left: co-monotonicity matrix, showing the value of $(\boldsymbol{u}_i - \boldsymbol{u}_j)^\top (Q(\boldsymbol{u}_i) - Q(\boldsymbol{u}_j))$ for all $i, j$, after fitting VQR without VMR with $d = 2$, $T = 10$ on the MVN dataset. Right: quantile-level pairs $i, j$ where co-monotonicity is violated. VMR resolves all these violations.

### F.4 NONLINEAR VQR

Here we include additional results comparing linear and nonlinear VQR for various datasets. Figure A12c compares linear and nonlinear VQR on the synthetic glasses dataset. Figures A13 and A14 and table A3 present the results of linear and nonlinear VQR on the conditional-banana and rotating star datasets for many values of $\boldsymbol{x}$.

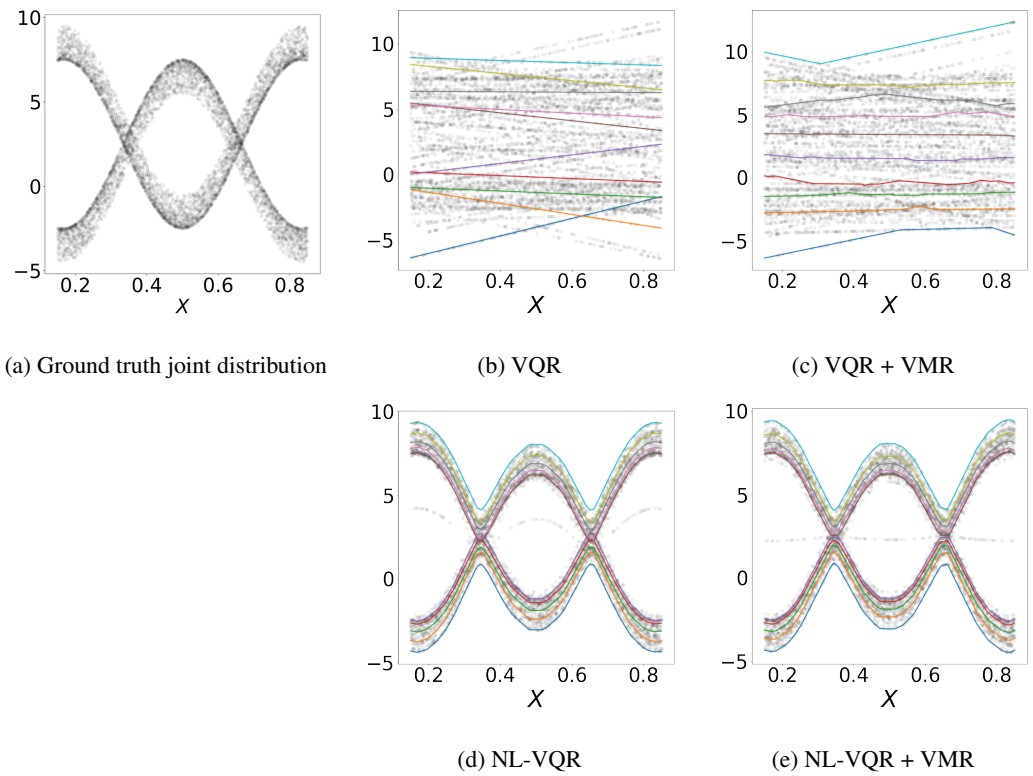

(a) Ground truth joint distribution      (b) VQR      (c) VQR + VMR

(d) NL-VQR      (e) NL-VQR + VMR

Figure A12: **VQR and NL-VQR can be used for Simultaneous Quantile Regression (SQR), and VMR eliminates quantile crossings.** SQR on the synthetic glasses dataset (a; gray points show ground-truth distribution). Linear VQR fails to correctly model the conditional distribution (b, c; gray points sampled from linear VQR), while nonlinear VQR reconstructs the shape exactly (d, e; gray points sampled from nonlinear VQR). In both cases case, VMR successfully eliminates any quantile crossings (c, e).

Table A2: **Quantitative evaluation of linear VQR, CVAE and nonlinear VQR on the conditional-banana dataset.** Evaluated on $N = 4000$ samples with $T = 50$ levels per dimension. The KDE-L1, QFD and Entropy metrics are defined in section 6. Arrows ↑/↓ indicate when higher/lower is better; Numbers are mean $\pm$ std calculated over the 20 values of $\boldsymbol{x}$. Reference entropy is obtained by uniformly sampling the 2d quantile grid.

|  |  | CVAE | VQR | NL-VQR |
|---|---|---|---|---|
| KDE-L1 | ↓ | $0.175 \pm 0.021$ | $0.873 \pm 0.175$ | $\mathbf{0.135} \pm 0.024$ |
| Quantile Function Distance (QFD) | ↓ | - | $0.179 \pm 0.034$ | $\mathbf{0.055} \pm 0.023$ |
| Inverse CVQF Entropy (Ref: 0.70) | ↑ | - | $0.309 \pm 0.077$ | $\mathbf{0.560} \pm 0.014$ |
| Train time (min) | ↓ | 35 | $\mathbf{3.5}$ | 4 |

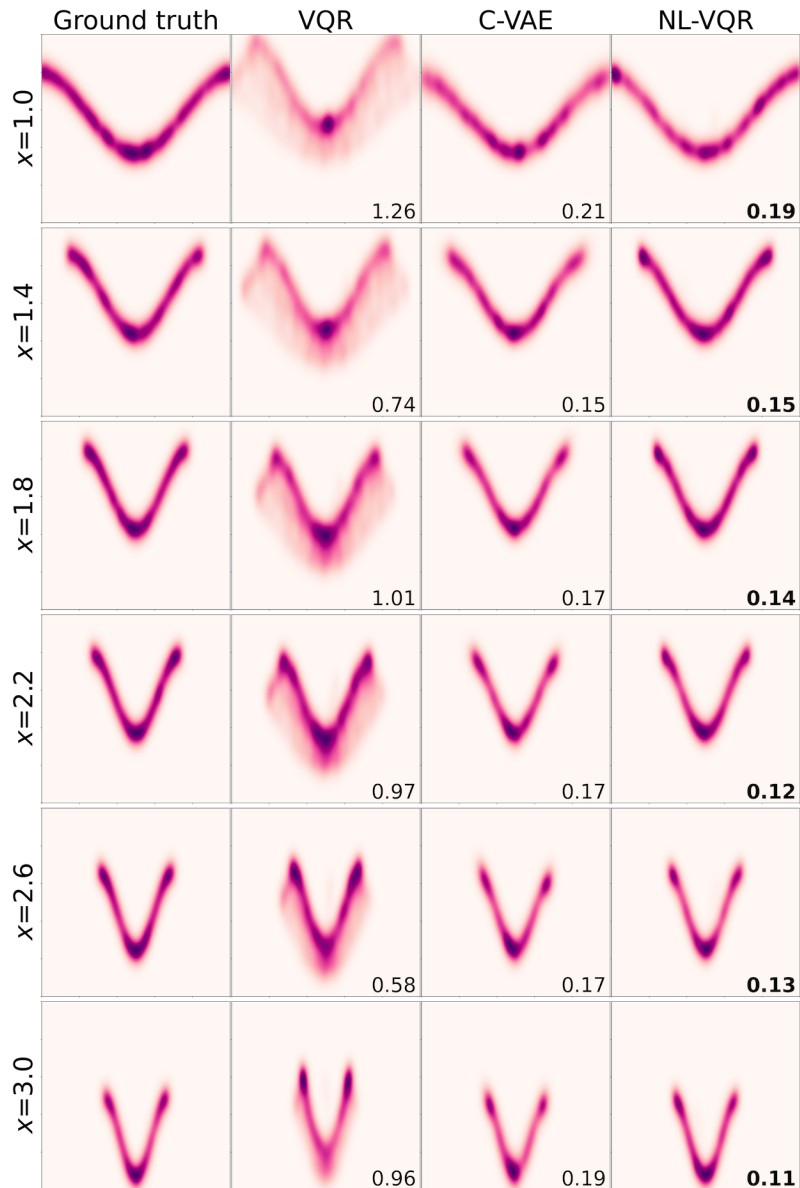

Figure A13: **Qualitative evaluation of VQR, CVAE and NL-VQR on the conditional-banana dataset.** Depicted are the kernel-density estimates of samples drawn from each of these methods. Models were trained with $N = 20k$ with $T = 50$ quantile levels per dimension. Numbers depict the KDE-L1 metric. Better viewed as GIF provided in the supplementary material.

Table A3: **Quantitative evaluation of linear VQR, CVAE, and nonlinear VQR on the rotating stars dataset.** Evaluated on $N = 4000$ samples with $T = 50$ levels per dimension. The KDE-L1, QFD and Entropy metrics are defined in section 6. Arrows ↑/↓ indicate when higher/lower is better; Numbers are mean $\pm$ std calculated over the 20 values of $x$. Reference entropy is obtained by uniformly sampling the 2d quantile grid.

|  |  | CVAE | VQR | NL-VQR |
|---|---|---|---|---|
| KDE-L1 | ↓ | $0.95 \pm 0.04$ | $0.35 \pm 0.06$ | $\mathbf{0.20} \pm 0.01$ |
| Quantile Function Distance (QFD) | ↓ | – | $0.18 \pm 0.04$ | $\mathbf{0.08} \pm 0.01$ |
| Inverse CVQF Entropy (Ref: 0.69) | ↑ | – | $0.51 \pm 0.02$ | $\mathbf{0.59} \pm 0.02$ |
| Train time (min) | ↓ | 35 | $\mathbf{3.5}$ | 4 |

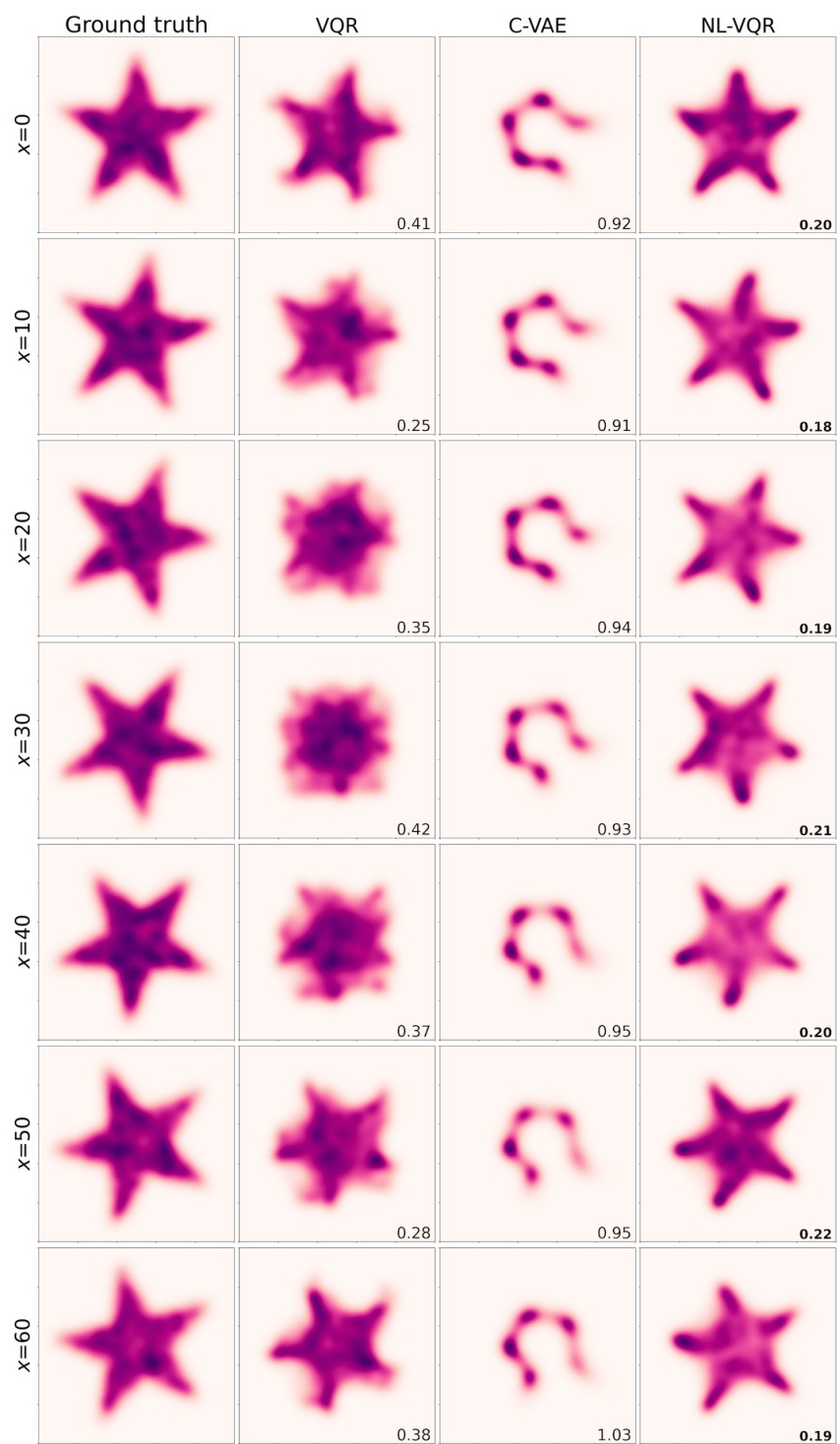

Figure A14: **Qualitative evaluation of linear VQR, CVAE and nonlinear VQR on the rotating stars dataset.** Depicted are the kernel-density estimates of samples drawn from each of these methods. Models were trained with $N = 20k$ with $T = 50$ quantile levels per dimension. Numbers depict the KDE-L1 metric. Better viewed as GIF provided in the supplementary material.

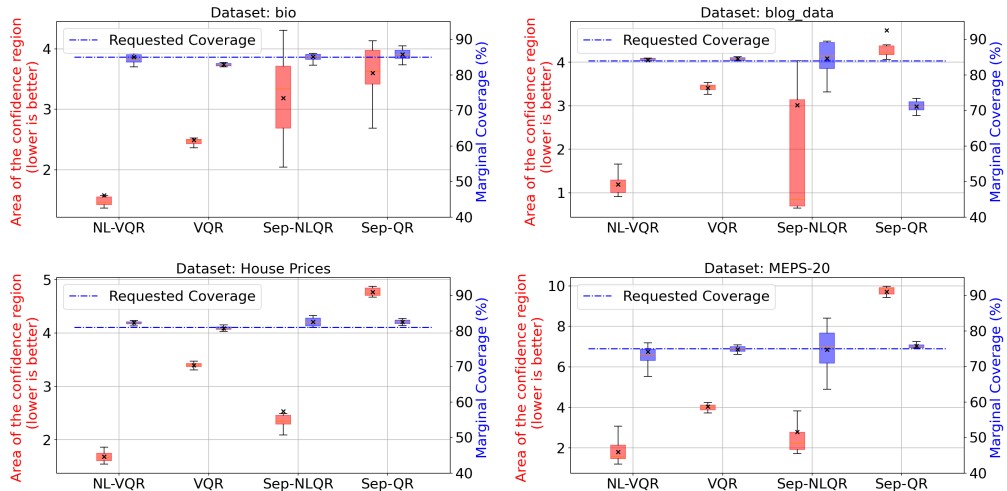

Figure A15: **Nonlinear VQR produces substantially smaller $\alpha$-confidence sets in real data experiments.** Comparison of nonlinear VQR (NL-VQR), separable nonlinear QR (Sep-NLQR), linear VQR (VQR), and linear separable QR approaches (Sep-QR) on the four real datasets presented in table A1. The confidence sets are constructed for each method in such a way to produce same desired level of marginal coverage (blue dashed line). Error bars generated from 10 trials, each with a different randomly-sampled train/test split.

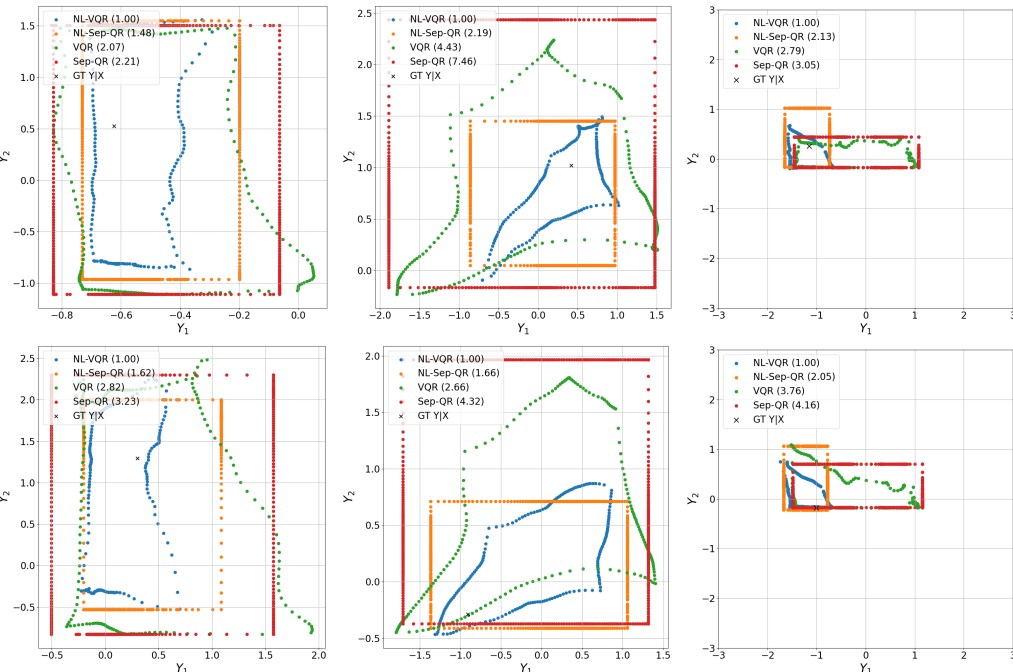

Figure A16: **Nonlinear VQR produces $\alpha$-quantile contours which model the shape of the conditional distribution and have significantly smaller area compared to other methods**. Each column depicts $\alpha$-quantile contours for two test covariates (top, bottom) sampled from `bio` (left), `house prices` (middle), and `blog_data` (right) datasets. Separable QR approaches produce box-shaped confidence regions due to the assumption of statistical independence, whereas VQR and nonlinear VQR produce contours with non-trivial shapes that adapt to the test covariate at hand (each column, top vs bottom). The areas of the $\alpha$-confidence sets are reported relative to NL-VQR in the legend.

## G   IMPLEMENTATION DETAILS

**Solver implementation details.**   The log-sum-exp term in eq. (7) becomes numerically unstable as $\varepsilon$ decreases, which can be mitigated by implementing it as

$$\mathrm{logsumexp}_\varepsilon(\boldsymbol{x}) = \mathrm{logsumexp}_\varepsilon\left(\boldsymbol{x} - \max_k\{x_k\}\right) + \max_k\{x_k\}.$$

**Scale and Optimization experiment.**   For both scale and optimization experiments, we run VQR for $10k$ iterations and use a learning rate scheduler that decays the learning rate by a factor of $0.9$ every $500$ iterations if the error does not drop by $0.5\%$.

**Synthetic glasses experiment.**   We set $N = 10k$, $T = 100$, and $\varepsilon = 0.001$. We optimized both VQR and NL-VQR for $40k$ iterations and use a learning rate scheduler that decays the learning rate by a factor of $0.9$ every $500$ iterations if the error does not drop by $0.5\%$. In NL-VQR, as $g_\theta$, we used a 3-layer fully-connected network with each hidden layer of size $1000$. We used skip-connections, batch-norm, and ReLU nonlinearities between the hidden layers. For NL-VQR and VQR, we set the initial learning rate to be $0.4$ and $1$, respectively.

**Conditional Banana and Rotating Star experiments.**   We draw $N = 20k$ samples from $P_{(\mathrm{X},\mathbf{Y})}$ and fit $T = 50$ quantile levels per dimension ($d = 2$) for linear and nonlinear VQR (NL-VQR). Evaluation is performed by measuring all aforementioned metrics on $4000$ evaluation samples from $20$ true conditional distributions, conditioned on $x = [1.1, 1.2, \ldots, 3.0]$. For NL-VQR, as $g_{\boldsymbol\theta}$ we use a small MLP with three hidden layers of size $(2, 10, 20)$ and a ReLU non-linearity. Thus $g_{\boldsymbol\theta}$ lifts X into 20 dimensions on which VQR is performed. We set $\varepsilon = 0.005$ and optimized both VQR and NL-VQR for $20k$ iterations. We used the same learning rate and schedulers as in the synthetic glasses experiment.

**Real data experiments.**   In all the real data experiments, we randomly split the data into $80\%$ training set and $20\%$ hold-out test set. Fitting is performed on the train split, and evaluation metrics, marginal coverage and quantile contour area (calculated as reported in appendix E.2), are measured on the test split. We repeat this procedure $10$ times with different random splits. The baselines we evaluate are NL-VQR, VQR, separable nonlinear QR (Sep-NLQR), and separable linear QR (Sep-QR). In the separable baselines, two QR models are fit separately, one for each target variable. For both nonlinear separable QR and NL-VQR baselines, we choose $g_\theta$ to be an MLP with three hidden layers. In the case of NL-VQR, hidden layers sizes are set to $(100, 60, 20)$ and for nonlinear separable QR, they are set to $(50, 30, 10)$. All methods were run for $40k$ iterations, with learning rate set to $0.3$ and $\varepsilon = 0.01$. We set $T = 50$ for NL-VQR and VQR baselines, and $T = 100$ for separable linear and nonlinear QR baselines. The $\alpha$ parameter that determines the $\alpha$-quantile contour construction is calibrated separately for each method and dataset. The goal of the calibration procedure is to achieve consistent coverage across baselines in order to allow for the comparison of the $\alpha$-quantile contour areas. Table A4 presents the calibrated $\alpha$'s that were used.

| Dataset | NL-VQR | | | VQR | | | Sep-NLQR | | | Sep-QR | | |
|---|---|---|---|---|---|---|---|---|---|---|---|---|
| | $\alpha$ | Cov. | Area | $\alpha$ | Cov. | Area | $\alpha$ | Cov. | Area | $\alpha$ | Cov. | Area |
| `bio` | 0.05 | 85.00 | 1.57 | 0.03 | 82.87 | 2.49 | 0.01 | 85.13 | 3.18 | 0.02 | 85.76 | 3.59 |
| `blog_data` | 0.05 | 84.17 | 1.19 | 0.05 | 84.50 | 3.41 | 0.025 | 84.60 | 3.02 | 0.025 | 71.10 | 4.73 |
| `house` | 0.05 | 82.29 | 1.69 | 0.06 | 80.75 | 3.39 | 0.025 | 82.51 | 2.53 | 0.03 | 82.48 | 4.76 |
| `meps_20` | 0.02 | 74.13 | 1.81 | 0.06 | 74.82 | 4.04 | 0.01 | 74.73 | 2.78 | 0.01 | 75.77 | 9.70 |

Table A4: **Choice of $\alpha$ for different real datasets.** Nominal (requested) coverage for `bio` and `blog_data` was $85\%$. For `house` and `meps_20` datasets, it was $82\%$ and $75\%$, respectively.

**Machine configuration.**   All experiments were run on a machine with an Intel Xeon E5 CPU, 256GB of RAM and an Nvidia Titan 2080Ti GPU with 11GB dedicated graphics memory.

# H SOFTWARE

We provide our VQR solvers as part of a comprehensive open-source python package `vqr` which installed with `pip install vqr`. The source code is available at `https://github.com/vistalab-technion/vqr` and it contains several `jupyter` notebooks which demonstrate the use of our package.

```python
import numpy as np
from vqr import VectorQuantileRegressor
from vqr.solvers.regularized_lse import RegularizedDualVQRSolver

# Create the VQR solver and regressor.
vqr_solver = RegularizedDualVQRSolver(
    verbose=True, T=50, epsilon=1e-2, num_epochs=1000, lr=0.9
)
vqr = VectorQuantileRegressor(solver=vqr_solver)

# Fit the model on some training data.
vqr.fit(X_train, Y_train)

# Sample from the fitted conditional distribution, given a specific x.
Y_sampled = vqr.sample(n=100, x=X_test[0])

# Calculate coverage on the samples.
cov_sampled = vqr.coverage(Y_sampled, x=X_test[0], alpha=alpha)
```

Listing 1: Minimal example of using the `vqr` library. Demonstrates instantiation of solver and regressor, fitting VQR, sampling from the fitted CVQF and coverage calculation.

We note that although a reference Matlab implementation of linear VQR by the original authors exists, this implementation relies on solving the exact formulation (5) using a general-purpose linear program solver. We found it to be prohibitively slow to use general-purpose linear program solvers for VQR (fig. A9). For example, we noted a runtime of 4.5 hours for $N = 7k, d = 2, k = 10, T = 30$, and for $N = 10k$ the solver did not converge even after 8 hours. Thus, this approach is unsuitable even for modestly-sized problems. Moreover, when using such solvers, only linear VQR can be performed. For comparison, our proposed solver converges to a solution of the same quality on problems of these sizes in less than one minute and supports nonlinear VQR.

