# OpenReview forum: "Fast Nonlinear Vector Quantile Regression"
_ICLR.cc/2023/Conference — ICLR 2023 poster_

### Official Review · Reviewer_u5QX · 2022-10-24

**Confidence:** 3
**Clarity, Quality, Novelty And Reproducibility:** See above.
**Correctness:** 3
**Technical Novelty And Significance:** 3
**Empirical Novelty And Significance:** 3
**Recommendation:** 8

**Strength And Weaknesses:**

Strengths:
- Clean abstract in terms of the limitations addressed and the included contributions.
- Good visualization/illustration of vector quantiles in Figure 1.
- Brief summary of previous work in introduction and discussion of related work throughout the paper.
- Even though I am not too familiar with this line of work, vector quantiles seem to be relevant in several niche applications and this paper provides a more scalable solver. With the increased interest in distribution-free uncertainty estimation, I can imagine this becoming and interesting tool.
- The extension to non-linear models seems to be a particularly important contribution given the experiments and I can imagine this to be impactful in this line of work.
- I appreciate the experiments on conformal prediction that show the benefit of this method.
- Providing an open-source implementation.

Weaknesses:
- The paper is generally more difficult to follow, mainly due to two reasons: the content being very technical and dense (requiring a lot of familiarity with the topic or time to go through equations by trial and error) and missing structure. For example, in section 3, there is a high-level outline/overview missing. The primal discussion, as example, seems less relevant and not be used (more like a comment on related work), while the dual formulation and SGD approach is actually used in the proposed method.
- The discussion of SQR and prior attempts in section 4, while interesting, seems less important in the main paper and distracted me. I’d prefer this space to be spent on more structure and a bit more guidance through the technical content.
- When less familiar with this line of work, the experimental setup could also use more details – e.g., describe at least one metric properly and give a high-level idea of the datasets.
- The details of the contribution over Genevay 2016 in terms of the solver is a bit unclear to me. As of my understanding the SGD optimization setup of that paper is used with the “trick” of obtaining constant memory footprint. Is that correct? This could be made clearer throughout the paper. In the abstract/intro it sounds like a novel solver is proposed instead.
- I couldn’t find the exact non-linear model that is used in the main paper and would be interested in a discussion of the constraints here. Are generally any differentiable non-linear models supported?
- I am also missing a baseline in terms of runtime. In Figure 2, how would previous solvers perform? When would they stop working?
- While I appreciate the conformal prediction experiments, this part is difficult to follow and judge from the main paper as the results are exclusively found in the appendix.
- Generally, this paper and the experiments are difficult to read without the appendix and I think this hurts the paper. I would appreciate if the authors could prioritize experiments and discuss a subset where the results are actually in the main paper, instead.

**Summary Of The Paper:**

The authors propose a scalable solver for non-linear multivariate quantile regression. While vector quantiles have been defined and studied in previous work, the authors address several limitations: developing a scalable solver, extending the linear regressor model to allow non-linear models and making sure monotonicity is ensured.

**Summary Of The Review:**

I believe this is an interesting contribution even though writing and structure could be improved significantly. Also, I am not too familiar with the background on vector quantiles, but appreciated the conformal prediction results.

---

> ### Author Response · Authors · 2022-11-14
> **Authors' Response (part 1)**
>
> We thank the reviewer for the appreciative review and constructive criticism.
>
> **Organization of the paper.**
> We thank the reviewer for suggesting constructive ways to improve the clarity of the paper. We have made changes based on the suggestions, which are detailed below. We also briefly explain here our rationale for the paper’s organization.
>
> The key challenge we encountered while writing this paper was that the notions of vector quantiles (VQs) and VQR are relatively unknown in the ML community. The tool we developed relies heavily on the domains of quantile regression (QR) and optimal transport (OT), and therefore in order for the reader to appreciate our contribution, we needed to familiarize them with both to some extent.
>
> We aimed to organize the paper such that the reader can understand the tool and consider potential applications for it, by only reading the main section. It was therefore important to provide a self-contained and brief introduction to OT-based QR, VQs and VQR along with an explanatory figure of these notions (Figure 1). Therefore, the main section of the paper contains comprehensive background, as the reviewer kindly acknowledged, which we deemed important for presenting the contributions in an accessible way. For example, including the primal formulation in the paper improves clarity for readers familiar with OT, and further helped us explain why the Sinkhorn algorithm is not applicable. It was also needed to motivate the proposed formulation which, as we explain, is equivalent to the entropic-regularized primal.
>
> We view this work as presenting a general purpose tool for estimating the quantiles of $\mathbf{Y}|\mathbf{X}$. As such, applications of this tool are of independent interest, and we decided to show only one example application of a problem where NL-VQR is the natural solution, namely distribution-free uncertainty quantification (DFUQ). We thank the reviewer for appreciating the results of this example. We agree that placing the result figures of the DFUQ experiments solely in the appendix hurts the paper. To address this remark, we reorganized the experiment section in the following way. We moved the optimization experiment results to the appendix, and moved two DFUQ result figures from the appendix to section 6.
>
> Furthermore, in order to improve the readability of the paper and following the reviewer’s suggestion, we added introductory sentences at the start of sections 2-5 which help guide the reader through the technical content. We also added more paragraph titles in these sections to better organize them.

---

> ### Author Response · Authors · 2022-11-14
> **Authors' Response (part 2)**
>
> **Choice of $g_{\theta}(x)$.**
> The exact nonlinear model we used is a simple 3-layer MLP described in Appendix G. Our implementation indeed allows the use of any differentiable model as $g_{\theta}(x)$.
>
> **Baseline for runtime.**
> We define a comparable baseline as solving linear VQR using the primal OT formulation with a general-purpose linear program solver. To fully address the reviewer’s remark, we conducted additional experiments and added a figure providing this comparison in the appendix (Fig. A9). To summarize, we found it to be prohibitively slow to use general-purpose linear program solvers for VQR. For example, we noted a runtime of 4.5 hours for $N=7k, d=2, k=10, T=30$, and for $N=10k$ the solver did not converge even after 8 hours. Moreover, when using such solvers, only linear VQR can be performed. For comparison, our proposed solver converges to a solution of the same quality on problems of these sizes in less than one minute and supports nonlinear VQR.
>
> **Contribution related to scalable VQR.**
> We thank the reviewer for pointing out this potential source of confusion. To clarify these points for the reader, we have made a few edits to the introduction distinguishing between the new formulation and the new solver.
>
> Genevay et al 2016 showed that it is possible to build scalable SGD-based solvers for the classical OT problem (Euclidean ground metric) using the regularized dual formulation. Our problem is not classical OT because it has a different ground metric and an additional mean-independence constraint that involves $\mathbf{X}$. Our approach is similar in spirit to Genevay et al. 2016, in the sense that we develop a new regularized dual formulation for VQR, that allows extension to nonlinear specification.
>
> Solving our proposed dual formulation with SGD presents a challenge, due to the large number of parameters that scale polynomially with the number of quantile levels $T$, which need to be kept in GPU memory during optimization. To obtain accurate estimation of the CVQFs one needs to use as high a value of $T$ as possible (Fig. 2), and when $\mathbf{X}$ is high-dimensional, it exacerbates the problem since batches of $\mathbf{X}$ need to be in memory as well.
>
> Our proposed solution is to use SGD with a custom modification for VQR. The quantile levels are randomly sampled together with the data samples. Then, as opposed to standard SGD where all model parameters are optimized per iteration, here only the CVQF model parameters corresponding to the sampled levels are optimized in each iteration.

---

### Official Review · Reviewer_fWcC · 2022-10-25

**Confidence:** 2
**Correctness:** 4
**Technical Novelty And Significance:** 4
**Empirical Novelty And Significance:** 4
**Recommendation:** 8

**Clarity, Quality, Novelty And Reproducibility:**

A large fraction of the paper, including the essential details of the experimental results, is tucked away in appendices. I feel like this might violate the spirit of the page limit. Perhaps a paper of this size is better suited as a journal article?

**Strength And Weaknesses:**

Strengths:
* If the authors' literature review is taken to be comprehensive, then they have advanced the state of the art for vector quantile regression (VQR) by quite a large amount: they introduce the first (partially) nonlinear model for VQR, they introduce a vector variation of monotonic rearrangement, and they provide an efficient and scalable implementation.

Weaknesses:
* If I understand Section 4 correctly, the model does not allow for interactions between x and u. This seems to imply that the model can only model situations where each conditional distribution over Y (for a given x value) is exactly the same as the others (for all other x values), up to an affine scale-and-shift transformation. This seems like an overly-restrictive model, but I guess prior work, using a fully linear model in both u and x, is even more restrictive.
* Vector quantile regression seems like quite a niche topic. In my experience in corporate research, even scalar-valued quantile regression is little-known and little-used.
* There doesn't seem to be a comparison to, or a mention of, Bayesian methods or generative modeling, which seems like a very natural fit for this problem.

**Summary Of The Paper:**

The paper introduces an efficient, nonlinear model for vector quantile regression (VQR). VQR is the problem of learning the quantiles of a multivariate response variable Y, conditioned on set of covariates x. The approach extends prior work by allowing for a nonlinear function of x and implementing vector rearrangement to guarantee monotonicity of simultaneous quantile estimates.

I am not familiar with Optimal Transport, so unfortunately, a fraction of the paper's key ideas were unintelligible to me.

**Summary Of The Review:**

I really like this topic and feel that it deserves more attention. Unfortunately, the audience for this paper would be relatively small. However, I believe it extends the state of the art quite a bit for this problem, and as such it probably deserves to be published.

---

> ### Author Response · Authors · 2022-11-14
> **Authors' Response**
>
> We thank the reviewer for the thoughtful and encouraging review.
>
> **Interactions between $x$ and $u$.**
> We acknowledge that this limitation indeed exists (in the Limitations we wrote that the “non linear transformation $g_{\theta}(x)$ is shared across quantile levels”). The reason we opted for this formulation is because the CVQF is required to be the gradient of a convex function in $u$. We observed that including non-linear interactions between $x$ and $u$, makes it harder in practice for the optimization procedure to obtain this convexity. This results in estimating infeasible (non co-monotonic) CVQFs. To address the reviewer’s remark, we clarified the sentence in the limitations to better explain this point.
>
> In practice we observe that this limitation is not overly restrictive, as demonstrated by our results which show the ability of NL-VQR to accurately model complex distributions with arbitrary shapes (star, cond-banana) or with bi-modality where the location of the modes shifts with $x$ (synthetic glasses). By employing $g_{\theta}(x)$, the resulting CVQF model is linear in $g_{\theta}(x)$ instead of in $x$, and thus potentially better-specified than a linear CVQF model (as the reviewer also noted).
>
> That said, we believe this is an exciting direction for future research. Particularly, by building upon recent advancements in the field of continuous optimal transport, one can use partial input-convex neural networks to model a monolithic $g_{\theta}(x, u)$ which allows non-linear interactions between $x$ and $u$, while being convex in $u$.
>
> **Comparison to Bayesian and generative models.**
> Our goal with this paper was to accurately estimate $Q_{Y|X}$ without making any distributional assumptions. This statistical quantity cannot be estimated with any other approach we are aware of.
>
> A byproduct of estimating CVQFs, is the ability to perform sampling from the conditional distribution (via the inversion transform, i.e. uniformly sampling a quantile level and applying the CVQF to it). However, sampling is not the key use-case that we are targeting. For example, to construct confidence sets, the ability to sample is by itself insufficient. As we have demonstrated, by using vector quantiles, one can construct confidence sets in high dimensions.
>
> Generative models and Bayesian models do not estimate CVQFs. Therefore, one can compare VQR to them only in terms of sampling quality. We indeed experimented with conditional variational autoencoders (an approximate Bayesian method), and we observe that conditional sampling via the CVQF is of better quality in practice than that of CVAE (See e.g. Appendix Figs. 13-14).
>
> **Organization of the paper.**
> We thank the reviewer for the comment. Following this remark and those of Reviewer u5QX, we edited the paper in a way that hopefully improves its readability. We bring in some experimental result figures on distribution-free uncertainty quantification (DFUQ) experiments to the main paper in section 6. We also added guiding introductory sentences in the technical sections.
>
> Our aim with this paper was to provide the ML community with a self-contained introduction to VQR and ready to use software, and we organized the paper to facilitate this goal. We kindly refer the reviewer to our response for reviewer u5QX, where we further explain our rationale for the paper’s organization.
>
> **VQR is a niche topic.**
> Although QR and VQR may be niche tools, the applications they serve are very general. Regression is widely employed in everyday ML tasks. For some applications (e.g. in image processing [1]) it is even quite popular to use the L1 regression loss which then actually performs quantile regression “in disguise”, since this estimates the conditional median. In every instance where regression is applied, VQR can now be used to estimate the conditional quantiles instead of merely the conditional mean. Arguably this provides immediate benefit for sensitive applications where regression outputs are used for decision making (e.g., causal inference). With our contribution, one may use the same neural nets which are currently employed in conditional mean estimation as the learned feature embedding, $g_{\theta}(x)$, and perform nonlinear VQR to obtain conditional quantiles.
>
> In line with the reviewer, we also believe that QR and VQR deserve much more recognition than they currently receive, given the value they provide. We therefore hope that VQR will become a standard part of the data scientist's toolbox. With this work we aim to serve two purposes for ML researchers and practitioners: (1) familiarize them with the notions of VQFs, CVQFs and their uses; (2) enable them immediately to apply VQR on their problems of interest as an off-the-shelf tool.
>
> [1] Zhao, Hang, et al. "Loss functions for image restoration with neural networks." IEEE Transactions on computational imaging 3.1 (2016): 47-57.

---

### Official Review · Reviewer_2xhV · 2022-10-26

**Confidence:** 3
**Correctness:** 3
**Technical Novelty And Significance:** 3
**Empirical Novelty And Significance:** 4
**Recommendation:** 8

**Clarity, Quality, Novelty And Reproducibility:**

This paper is generally well-written, and it seems to have sufficiently novel and significant contributions that may advance the field.

The code has been included in the supplementary material (and also in GitHub) for reproducibility.

**Details Of Ethics Concerns:**

NA.

**Strength And Weaknesses:**

Strength:

The studied problem is of sufficient interest to the community, and the authors' work seems to be a significant contribution to the field of VQR.

Weaknesses:

Although the experimental results are sufficiently promising, I hope that similar to some closely relevant works such as Carlier et al. (2016) and Chernozhukov et al. (2017), the authors can provide some theoretical justifications for their proposed methods.

**Summary Of The Paper:**

In this work, motivated by the limitation that there are no fast or scalable solvers for vector quantile regression (VQR), the authors provide a highly-scalable solver for VQR that relies on solving its relaxed dual formulation. In addition, the authors propose vector monotone rearrangement (VMR), which serves as a refinement step and resolves the co-monotonicity violations in estimated conditional vector quantile functions (CVQFs). Moreover, motivated by the limitation of VQR that it assumes a linear model for the quantiles of the target given the features, the authors propose nonlinear vector quantile regression (NL-VQR). Based on extensive experiments on both synthetic and real data, the authors demonstrate the superiority of VMR and NL-VQR in modeling CVQFs accurately.

**Summary Of The Review:**

Overall, I think that this is a good work with sufficiently novel methods and interesting numerical results.

---

> ### Author Response · Authors · 2022-11-14
> **Authors' Response**
>
> We thank the reviewer for the positive review, for acknowledging the importance of VQR, and for appreciating our contributions.
>
> **Theoretical justifications.**
> The existence and uniqueness of CVQFs in their most general sense (without assuming a model) was shown by Carlier et al (2016). They further assume a linear model only to obtain a tractable optimization problem. Our work builds upon their result and increases the representation power of the assumed model, thereby decreasing the approximation error of the resulting estimated CVQF with respect to the true CVQF. We believe these points are mentioned in sections 1 and 4, however we would be happy to clarify it further if the reviewer finds it unclear from reading the paper.
>
> The formulation we use for obtaining estimated CVQFs relies on the relaxed dual formulation presented in equation (8). This formulation produces an estimate $\widehat{Q}^{\varepsilon}$. It can be shown (by following the approach of Proposition A.1 in Genevay et al 2016) that this estimate approaches the true CVQF as $\varepsilon \to 0$. Following the reviewer's remark, we edited Appendix B to add this justification.
>
> In the context of VMR, we have provided theoretical justifications as to why the procedure ensures co-monotonicity while strictly improving the estimation error of the CVQF (section 5).
>
> We also included a brief discussion in Appendix B regarding the justification for the use of SGD in the case of linear and nonlinear VQR.

---

### Decision · Program_Chairs · 2023-01-20

**Decision:**

Accept: poster

**Justification For Why Not Higher Score:**

The reviewers did not recommend a spotlight or above for this paper.
I am not confident enough to suggest a higher rating, especially since
none of the reviewers has strong expertise on the topic.

**Justification For Why Not Lower Score:**

My justification is based on the uniformly positive scores and my reading of the reviews.

**Metareview: Summary, Strengths And Weaknesses:**

This work studies a natural formulation of the multivariate quantile regression task.
Specifically, the authors defined a non-linear version of the problem and
presented an efficient method to solve it, including experimental results.
The reviewers agreed that this is a solid contribution that should be accepted.

**Note From Pc:**

if the above contains the word "oral" or "spotlight" please see: "oral" presentation means -> notable-top-5% and "spotlight" means -> notable-top-25%. As stated in our emails, we are disassociating presentation type from AC recommendations